# 🤖 Multi-modal Agent Tuning: Building a VLM-Driven Agent for Efficient Tool Usage

**Zhi Gao**[1,2*], **Bofei Zhang**[2*], **Pengxiang Li**[3,2*], **Xiaojian Ma**[2], **Tao Yuan**[2], **Yue Fan**[2]
**Yuwei Wu**[3,4✉], **Yunde Jia**[4], **Song-Chun Zhu**[1,2,5], **Qing Li**[2✉]

[1]School of Intelligence Science and Technology, Peking University
[2]State Key Laboratory of General Artificial Intelligence, BIGAI
[3]Beijing Key Laboratory of Intelligent Information Technology, School of Computer Science & Technology, Beijing Institute of Technology
[4]Guangdong Laboratory of Machine Perception and Intelligent Computing, Shenzhen MSU-BIT University
[5]Department of Automation, Tsinghua University
mat-agent.github.io

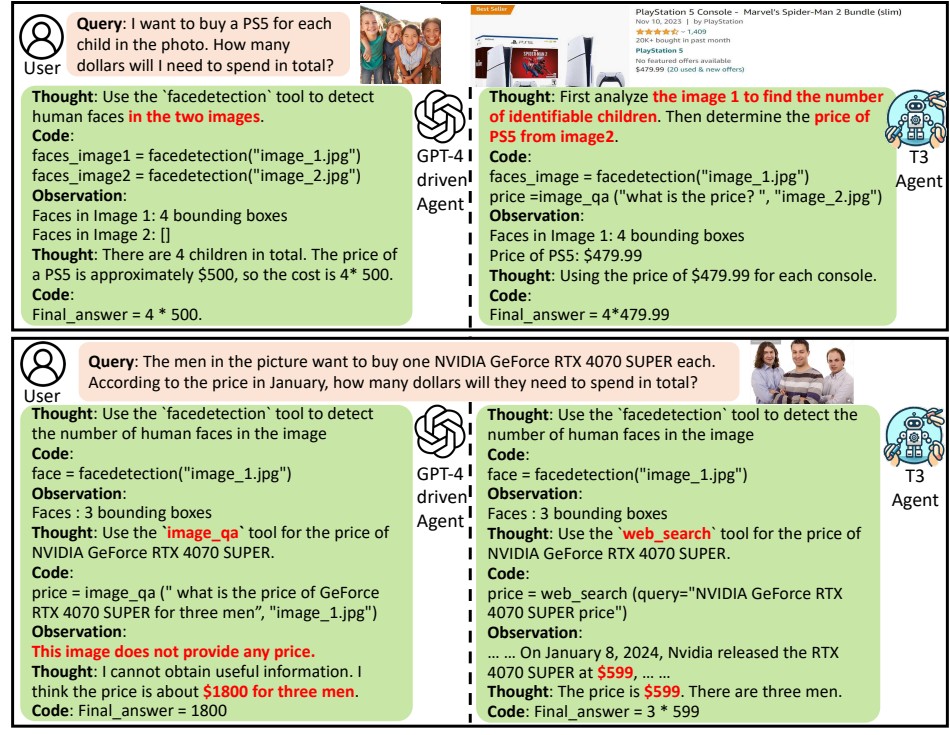

Figure 1: The comparison of the LLM (GPT-4)-driven agent and our T3-Agent. Our agent chooses more precise tools based on the given files and intermediate observations.

## Abstract

The advancement of large language models (LLMs) prompts the development of multi-modal agents, which are used as a controller to call external tools, providing a feasible way to solve practical tasks. In this paper, we propose a multi-modal agent tuning method that automatically generates multi-modal tool-usage data and tunes a vision-language model (VLM) as the controller for powerful tool-usage reasoning. To preserve the data quality, we prompt the GPT-4o mini model to generate queries, files, and trajectories, followed by query-file and trajectory verifiers. Based on the data synthesis pipeline, we collect the MM-Traj dataset that contains 20K tasks with trajectories of tool usage. Then, we develop the T3-Agent via Trajectory Tuning on VLMs for Tool usage using MM-Traj. Evaluations on the GTA and GAIA benchmarks show that the T3-Agent consistently achieves improvements on two popular VLMs: MiniCPM-V-8.5B and Qwen2-VL-7B, which outperforms untrained VLMs by 20%, showing the effectiveness of the proposed data synthesis pipeline, leading to high-quality data for tool-usage capabilities.

---

[*]Equal contribution.  ✉ Corresponding author.

# 1 INTRODUCTION

Integrating external tools to solve diverse multi-modal tasks is a promising research direction towards multi-modal agents (Surís et al., 2023; Gupta & Kembhavi, 2023; Gao et al., 2024; Yuan et al., 2024; Zhong et al., 2023). Existing agents usually use a large language model (LLM) as the controller that generates plans via prompt engineering to call tools, achieving impressive performance in multiple domains, such as image editing (Wu et al., 2023), robotic manipulation (ichter et al., 2023), question answering (Shen et al., 2024), video understanding (Fan et al., 2024), and desktop APPs (Trivedi et al., 2024). Despite their success, prompt engineering faces limited reasoning abilities for tool usage in tackling practical tasks, as shown in Fig. 1. (1) The in-context examples in prompts only involve textual information, degrading the efficiency of tool usage in the multi-modal world. For the query 'How many dollars will I need to spend to buy a PS5 for each child?', the agent may select improper tools if it does not know what the two images depict. (2) The pre-defined in-context examples are fixed and cannot tackle all tasks in the real world. For the task that requires searching for information from the web, the agent cannot use the proper tools, if in-context examples tend to use the 'image_qa' tool. This motivates us to enhance the controller's reasoning ability for efficient tool usage.

In this paper, we propose a multi-modal agent tuning method that automatically generates a large number of multi-modal tasks with tool-usage trajectories and tunes a vision-language model (VLM) (Liu et al., 2024b; Chen et al., 2024d; Yao et al., 2024) as the controller for powerful tool-usage reasoning. Compared with LLM-driven agents, VLM-driven agents can utilize multi-modal information (such as required knowledge domains in the multi-modal data) instead of only using the query for reasoning, benefiting efficient tool usage (Liu et al., 2024c; Wang et al., 2024a; Sun et al., 2024). Many efforts are made to enhance specific capabilities of VLMs via finetuning, such as the chain-of-thought ability (Hu et al., 2024), grounding ability (Peng et al., 2023), and feedback-refining ability (Li et al., 2024). This inspires us to construct a large number of multi-modal tool-usage data for VLM-driven agents, which improves the reasoning ability when using tools for real-world tasks.

In doing so, we need to overcome two challenges. (1) Collecting multi-modal tasks is challenging. Tasks in the real world usually involve multiple tools for multiple files (images, textual files, videos, audio, and *etc*). There are few off-the-shelf datasets for such tasks, and prompting models to generate natural and diverse queries with matched files is non-trivial. (2) Generating trajectories is challenging. Due to the complexity of trajectories, existing methods usually manually define templates and fill in key information for trajectory generation. This will limit the diversity of synthesis data and cause weak generalization for real-world tasks.

To overcome the above challenges, we introduce a novel tool-usage data synthesis pipeline that automatically generates a large number of multi-modal tool-usage data via three steps: query generation, file generation, and trajectory generation. Concretely, we first prompt GPT-4o mini (OpenAI, 2024) to generate queries and analyze what files are needed to solve the queries. Then, we produce files via two strategies. If needed files are images, we search for them from existing image datasets; otherwise, we prompt GPT-4o mini to produce codes to generate the needed files. Finally, we prompt a zero-shot agent to solve the generated tasks (*i.e.*, queries and files) and collect trajectories, including the thoughts and codes in task solving. To preserve the data quality, the generated tasks and trajectories are passed through two verifiers to discard low-quality data. After that, we use these data to tune a VLM for efficient tool usage, through which one agent driven by the trained VLM could generate precise thoughts and codes for real-world tasks.

With the data generation pipeline, we construct MM-Traj, a dataset that contains 20K multi-modal tasks with tool-usage trajectories. Based on MM-Traj, we introduce the T3-Agent, a VLM-driven agent in the ReAct framework (Yao et al., 2023). The VLM controller of the T3-Agent is developed via Trajectory Tuning for Tool usage using MM-Traj. We conduct comprehensive evaluations of the T3-Agent on the GTA (Wang et al., 2024b) and GAIA benchmarks (Mialon et al., 2023), where two popular VLMs are used as the controller, that is MiniCPM-V-8.5B (Yao et al., 2024) and Qwen-VL-7B (Wang et al., 2024c). The T3-Agent consistently achieves improvements on the two VLMs and outperforms the untrained VLMs by $20\%$. This indicates that our multi-modal agent tuning method enables agents a powerful tool-usage capability for complex and diverse trajectories.

In summary, our contributions are three-fold. (1) We propose a multi-modal agent tuning method that automatically generates a large number of multi-modal tasks with trajectories and tunes VLMs using the generated data for powerful tool usage. (2) We introduce MM-Traj, a multi-modal tool-usage dataset that contains 20K tasks across diverse knowledge domains with 15K files and high-quality

trajectories. (3) We develop the T3-Agent, a multi-modal tool-usage agent that significantly improves the tool usage performance on two popular benchmarks: GTA and GAIA.

## 2 RELATED WORK

### 2.1 MULTI-MODAL AGENT

Using external tools to address complex tasks is an important ability for multi-modal agents. According to different controllers, existing agents can be categorized into LLM-driven agents and VLM-driven agents. LLM-driven agents utilize powerful LLMs as the controller and produce pseudo code (Gupta & Kembhavi, 2023; Gao et al., 2024), python code (Surís et al., 2023; Yuan et al., 2024), or JSON format (Shen et al., 2024) to call tools via one-step reasoning. Considering the complexity of practical tasks, some methods (Yang et al., 2023; Fan et al., 2024; Yang et al., 2024) empower the agent with step-by-step reasoning, which allocates tools based on observations of previous steps. Compared with LLM-driven agents, VLM-driven agents are more efficient in task solving, since the VLM controller can utilize information from visual data in tool usage, showing superior performance in visual design (Sasazawa & Sogawa, 2024), web search (Zheng et al., 2024a), image editing (Wang et al., 2024e), embodied scenario (Zheng et al., 2024b), robotic manipulation (Sun et al., 2024), and *etc*. However, VLM-driven agents have a weaker reasoning ability, compared with LLM-driven agents. Thus, several works synthesize training data to tune open-source VLMs for general tool usage (Wang et al., 2024a; Liu et al., 2023a; 2024c). Due to the challenges of trajectory generation, existing methods mainly focus on simple tasks requiring one or two tools, and only synthesize a small amount of data (*e.g.*, 1K in (Liu et al., 2024c)). Different from them, our T3-Agent is tuned using scaled-up multi-modal data with complex tool-usage trajectories, through which our agent could solve more practical tasks with strong tool-usage capability.

### 2.2 TOOL-USAGE DATASET

Several tool-usage datasets have been established for agents, such as APIBank (Li et al., 2023), Toolalpaca (Tang et al., 2023), ToolBench (Qin et al., 2023), AnyTool (Du et al., 2024), agentohana (Zhang et al., 2024a), APIGen (Liu et al., 2024d), and AgentInstruct (Zeng et al., 2023). The above datasets contain little multi-modal data (*e.g.*, images, videos, and audio) that are commonly encountered in the real world. Thus, to evaluate the performance of agents in solving multi-modal tasks, some multi-modal agent benchmarks have been built, such as the GUI benchmarks: OSWorld (Xie et al., 2024) and MMInA (Zhang et al., 2024b), multi-modal question answering benchmarks: GAIA (Mialon et al., 2023), GTA (Wang et al., 2024b), and m&m' (Ma et al., 2024), and comprehensive benchmarks: AgentBench (Liu et al., 2023b) and AgentGym (Xi et al., 2024). In addition, some efforts are paid to synthesize trajectory data using LLMs to improve the tool-usage ability of multi-modal agents. DEDER (Choi et al., 2024) resorts to in-context learning to generate trajectories, through which the chain-of-thought reasoning ability is distilled from LLMs to a small model. Lumos (Yin et al., 2024) converts ground-truth reasoning steps in existing benchmarks into the expected format of tool-usage trajectories. TASKBENCH (Shen et al., 2023) samples trajectories from a predefined graph, and then generate queries. MLLM-Tool (Wang et al., 2024a) and LLaVA-plus (Liu et al., 2023a) collect trajectories based on image and tool descriptions. VisualAgentBench (Liu et al., 2023b) manually designs trajectory templates to collect data. Different from the above methods that usually focus on simple and predefined trajectories, or rely on queries in off-the-shelf datasets, our data collection pipeline generates diverse tasks and complex trajectories without any constraints, improving the volume, complexities, naturalness, and diversity of the tool-usage dataset.

## 3 DATA COLLECTION

### 3.1 FORMULATION

**Data Synthesis Pipeline.** The proposed data synthesis pipeline is shown in Fig. 2, including three steps: query generation, file generation, and trajectory generation. To preserve the quality of data, we design a query-file verifier and a trajectory verifier to discard low-quality data.

**Data format.** We format the multi-modal tool-usage data as $\{F_{\text{opt}}, Q, T, C, O, A\}$, where $F_{\text{opt}}$ denotes the multi-modal files, $Q$ means the query, $T$ mean the generated thought (*i.e.*, plans to call tools), and $C$ means the generated code, $O$ means observation (outputs of using tools), and $A$ means the ground truth answer. $_{\text{opt}}$ means that the files are optional, *i.e.*, some queries do not involve files.

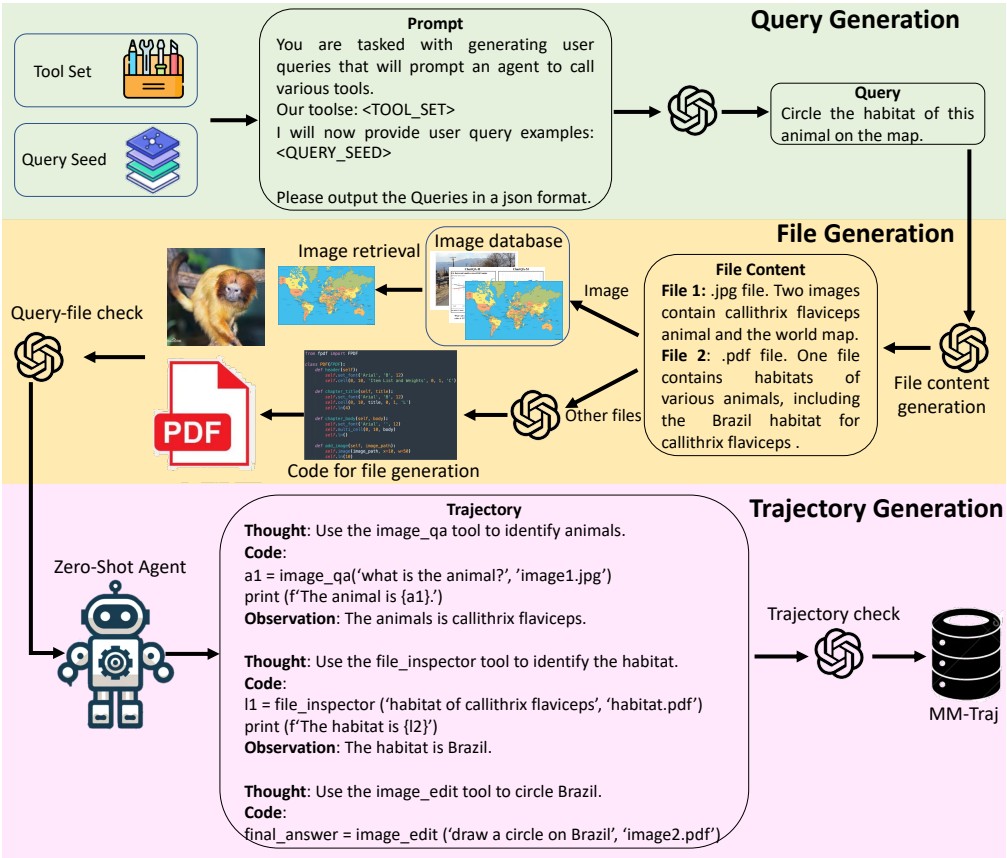

Figure 2: The pipeline for data generation.

The query $Q$ could be divided into two categories, question answering and image generation, where the answer $A$ would be some descriptive texts for the former and images for the latter. In our setting, $F$ includes 11 types of files, such as JPG, PDF and PPTX files (details in Appendix). Considering that solving one real-world task may require multiple steps involving multiple tools, $T$, $C$, and $O$ can be represented by the integration of thought, code, and observation in multiple steps, and the data format is reformulated as $\{F_{\text{opt}}, Q, \{t_1, \cdots, t_n\}, \{c_1, \cdots, c_n\}, \{o_1, \cdots, o_n\}, A\}$, where $t_i$, $c_i$, and $o_i$ indicate the thought, code, and observation in the $i$-step respectively, and there are $n$ steps in total. The thought, code, and observation are composed of a trajectory $\{t_1, c_1, o_1, \cdots, t_n, c_n, o_n\}$ of $n$ steps to solve the query.

**Image source.** Visual information plays an important role in multi-modal tasks. To enhance the diversity and comprehensiveness of our tool-usage data, we compile about 93K image-captioning pairs from 8 source datasets, including ChartQA (Masry et al., 2022), COCO (Lin et al., 2014), LLaVA (Wang et al., 2023a), SAM (Kirillov et al., 2023), TextVQA (Singh et al., 2019), Web-Celebrity (Liu et al., 2015), Web-Landmark (Weyand et al., 2020), and WikiArt (Saleh & Elgammal, 2015). These datasets cover multiple multi-modal tasks: visual question answering, chart/table/document analysis, science question answering, and *etc*. We use the ShareGPT4V model (Chen et al., 2024b) to produce captions for each collected image.

## 3.2 QUERY GENERATION

Our goal is to generate a large number of diverse, practical, and feasible queries. We manually write some seed queries by brainstorming and double-checking them to ensure their practicality. In each step of query generation, we feed several randomly sampled seed queries, tools with descriptions, and a designed prompt to the GPT-4o mini model that generates multiple queries. Adding tool descriptions to the prompt makes GPT-4o mini better understand the desirable queries, improving the feasibility of generated queries. We tune the hyperparameters (such as temperature) of GPT-4o mini to improve the diversity of generation.

### 3.3 MULTI-MODAL FILE GENERATION

Different from existing multi-modal data synthesis methods that first sample multi-modal files (images in most cases) from off-the-shelf datasets and then feed the files to language models (*e.g.*, ChatGPT) for query generation, we opt to first generate queries without files and then produce relevant files for the queries. The reasons are two aspects. (1) Practical tasks usually involve not only images but also other multi-modal files, such as DOCX, PPTX, XLSX, and PDF files. It is challenging to construct off-the-shelf datasets that contain enough files for real-world tasks. (2) Tasks are usually based on multiple files instead of only one, while randomly sampled files may have weak relevance, and feeding them to language models usually produces non-natural queries. It is non-trivial to design a standard to automatically sample relevant files for query generation. (3) Using existing files to generate queries may limit the knowledge domain, decreasing the diversity of tasks. In contrast, generating files based on generated queries may lead to more diversity.

Concretely, for each generated query, we prompt GPT-4o mini to output the needed file type and file content. The files are divided into two categories: images and others. For needed images, we use the BGE model (Chen et al., 2024a) to extract textual embeddings of the file content and compare its similarities with collected source images from off-the-shelf datasets. The top similar images are collected for the query. For other needed files, we prompt GPT-4o mini to extend the file content and generate Python code to produce the files.

### 3.4 ZERO-SHOT AGENT INFERENCE

Trajectories are collected by prompting a zero-shot agent (without training) to solve our generated tasks (*i.e.*, queries and files). We utilize the framework of ReAct agents (Yao et al., 2023), where GPT-4o mini is employed as the controller. It solves the query into multiple steps and generates thought and code for each step based on the observations of previous steps. We collect trajectories whose code can be executed, including the thoughts, codes, and observations of all steps. The details of the agent can be found in Section Appendix C and Appendix D.

### 3.5 DATA VERIFICATION

To preserve the quality of generated tool-usage data, we design a query-file verifier and a trajectory verifier to discard low-quality data. Using LLMs to verify the quality of LLM outputs has shown effectiveness in multiple methods, such as verifying the generated instructions (Wang et al., 2023b) and verifying the generated plans (Liu et al., 2024d). Inspired by them, we argue that using LLMs can also verify the synthetic tasks and trajectories.

**Query-file verifier.** Considering that the generated queries may be infeasible to solve and the produced files may not match the queries, the query-file verifier filters out low-quality query-file pairs. We prompt GPT-4o mini as the verifier based on the following factors: (1) whether the query and files are relevant; (2) whether the files contain enough information to solve the query; and (3) whether the query can be solved based on the given tools.

**Trajectory verifier.** Similarly, we prompt GPT-4o mini as the trajectory verifier based on the following factors: (1) the trajectory should use the provided tools as much as possible; (2) the trajectory should be reasonable, that is, the trajectory should align with the object and context of the query; (3) the tool usage in the trajectory should consistent with the query and files; (4) the input arguments for tools in the trajectory should be correct; (5) the answer should be correctly summarized from observations of tool-usage; and (6) the final answer should be relevant to the query.

### 3.6 MM-TRAJ DATASET

Data that passes through the two verifiers is considered high-quality and collected in an MM-Traj dataset. In summary, we collect 23.5K data points from query generation and file generation. After passing through the two verifiers, 20K data points are left with 15K files.

**Scalability.** Note that our method can extend to additional modalities by incorporating more tools and leveraging advanced multi-modal models. For example, to extend our method to the video modality (MP4, MOV), we can integrate a video search model into the data synthesis pipeline (like the image modality), and apply a video-language model to the agent controller with powerful video processing models as tools. This approach ensures seamless adaptation to new modalities while maintaining efficiency and coherence.

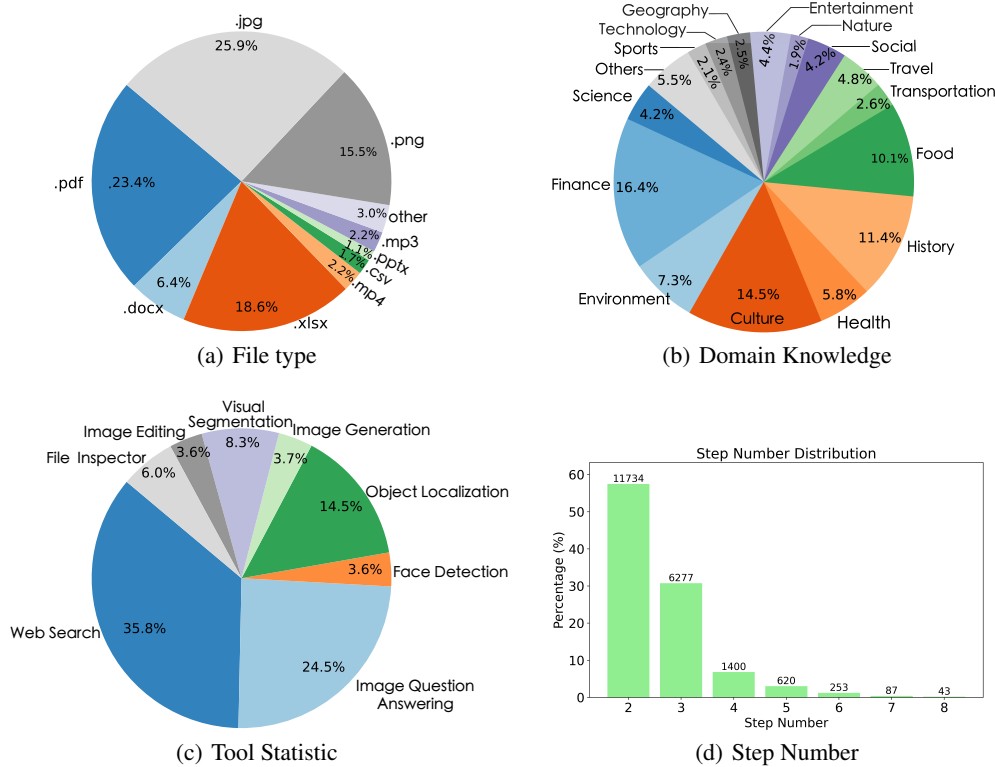

Figure 3: Data statistics on the MM-Traj dataset.

### 3.6.1 DATASET ANALYSIS

We provide four key statistics: file type, knowledge domain, step number, and used tools in the collected MM-Traj Dataset. **File type.** We show the distribution of files in Fig. 3(a), which reflects the diversity of our dataset. MM-Traj covers more than 9 kinds of files, all of which are commonly encountered in the real world. Thus, the T3-Agent trained on MM-Traj can handle practical tasks since the multi-modal knowledge is not limited to images in our lives. **Knowledge domain.** We show the involved knowledge of the generated tasks in Fig. 3(b), which can be divided into 16 non-overlap categories, spanning across finance, environment, culture, health, history, food, and *etc.* Training in such data provides rich knowledge to use tools to solve diverse practical tasks. **Tools.** In Fig. 3(c), we show the distributions of used tools in generated trajectories. The web search tool is the most commonly used tool, which is consistent with practical tasks requiring specific knowledge. Moreover, other tools are also widely used in our dataset. **Step number.** We show the distributions of step numbers of generated trajectories in Fig. 3(d). Trajectories in the MM-Traj dataset have diverse step numbers. Most tasks require 2-6 steps to solve, and some tasks require 7-8 steps, showing the complexity and diversity of our dataset.

## 4 T3-AGENT

### 4.1 WORKFLOW

To handle practical tasks that require complex trajectories, we opt for the framework of the ReAct agent that performs step-by-step reasoning for tool usage based on observations of previous steps. In each step, the agent generates thought and corresponding code to execute tools. Compared with other formats (*e.g.*, JSON format), code is more flexible to handle different inputs and output types for various tools. Concretely, given a query $Q$ and files $F$, the $i$-step of the agent is formulated as

$$t_i^\star, c_i^\star = \arg\max P(t_i, c_i | F_{\text{opt}}, T, Q, h_i), \tag{1}$$

where $t_i^\star$ and $c_i^\star$ are thought and code for the $i$-th step, and $h_i = \{t_1, c_1, o_1, \cdots, t_{i-1}, c_{i-1}, o_{i-1}\}$ denotes the history (thought, code, and observation of previous steps).

Table 1: Used tools in the T3-Agent

| Tool name | Tool description |
|---|---|
| Web search | Perform complicated web browsing to answer a question |
| Image question answering | Answer questions for queries based on attached images |
| File inspector | Answer questions for queries based on given files |
| Visual segmentation | Do instance segmentation on the given image |
| Object localization | Localize objects in given images and output the bounding boxes |
| Image generation | Create an image according to a textual prompt |
| Image editing | Edit image based on the textual prompt |
| Face detection | Detect human faces in given images and output the bounding boxes |
| Python package | Various packages such as 'matplotlib' and 'opencv' |

## 4.2 TOOLS

We deploy real-executable tools for the agent instead of only providing tool names. Our tools are across multiple categories: web search, visual perception, image generation/editing, file understanding, multi-modal understanding, and multiple Python packages, as shown in Tab. 1.

## 4.3 TRAINING

Given a data point $\{F_{opt}, Q, \{t_1, \cdots, t_n\}, \{c_1, \cdots, c_n\}, \{o_1, \cdots, o_n\}, A\}$, we train the VLM controller using the cross-entropy loss,

$$\min \mathbb{E}_{(F_{opt}, Q, T, C, O, A) \sim \mathbb{D}} \left[ -\sum_{i=1}^{n} P(t_i, c_i | F_{opt}, T, Q, h_i) \right], \tag{2}$$

where $\mathbb{D}$ is the collected MM-Traj dataset and we sum the loss values of the $n$ steps in the trajectory. Note that, in training VLMs, we do not use the final answer $A$, as we encourage the controller to leverage tools in solving given tasks, instead of directly producing an answer based on its internal knowledge. After training, we obtain the T3-Agent.

**Model.** We use the same model architectures as MiniCPM-V-8.5B (Yao et al., 2024) and Qwen2-VL-7B (Wang et al., 2024c) as our VLM controllers, including their visual encoders, resamplers, and LLMs. We initialize the model using their released versions.

## 5 EXPERIMENTS

### 5.1 SETTING

To evaluate the effectiveness of the proposed multi-modal agent tuning method, we evaluate the T3-Agent on the GTA (Wang et al., 2024b) and GAIA (Mialon et al., 2023) benchmarks and compare it with agents that use closed-source models (GPT-4, GPT-4o, and Claude3) and open-source models (LLaMA-3-70B-instruct (Dubey et al., 2024), Qwen1.5-72B-chat (Bai et al., 2023), LLaVA-NeXT-8B (Liu et al., 2024a), InternVL2-8B (Chen et al., 2024c), Qwen2-VL-7B (Wang et al., 2024c), and MiniCPM-V-8.5B (Yao et al., 2024)) as the controllers. Concretely, we compare the T3-Agent with Lego Agent (AgentLego Contributors, 2023), Sibyl Agent (Wang et al., 2024d), and the Warm-up Act Agent (Mialon et al., 2023). The huggingface agent (HF Agent) (HuggingFace Contributors, 2024) is the baseline agent, using the same tools as the T3-Agent. We conduct ablation experiments to evaluate our data synthesis pipeline and visualize the task-solving process of our T3-Agent.

**Training.** To preserve the visual perception and reasoning capabilities of MiniCPM-V and Qwen2-VL, we combine the training data in MM-Traj with the data in Cauldron (Lindström & Abraham, 2022) and open-LLaVa-NeXT (Chen, 2024) datasets. We train 5 epoch over all data. In the training process of our VLM controller, we freeze the vision encoder and visual token compressor, and fine-tune the language model using LoRA (Hu et al., 2022). We set the rank as 64 and apply LoRA on query, key, and value projection matrices in all self-attention layers. We use the AdamW optimizer with a cosine annealing scheduler. The learning rate is $1e - 6$ and the batch size is 2. We set the max context window to 10240 to support the long trajectory of our agent.

**Benchmark.** GTA and GAIA benchmarks are comprehensive evaluation benchmarks for multi-modal agents. GTA contains 229 tasks with 252 images, and the steps required to solve tasks range from 2 to 8, with most questions requiring 2 to 4 steps. It requires multi-modal agents to build powerful

Table 2: Results on the GTA benchmark

| Method | Controller | AnsAcc | ToolAcc | CodeExec |
|---|---|---|---|---|
| Lego Agent | GPT-4 | 46.59 | - | - |
| Lego Agent | GPT-4o | 41.52 | - | - |
| Lego Agent | GPT-3.5-turbo | 23.62 | - | - |
| Lego Agent | Claude3-opus | 23.44 | - | - |
| Lego Agent | Qwen1.5-72B-chat | 13.32 | - | - |
| Lego Agent | LLaMA3-70B-instruct | 8.32 | - | - |
| HF Agent | GPT-4o | 57.05 | 63.41 | 95.12 |
| HF Agent | GPT-4o mini | 57.69 | 56.10 | 100.00 |
| HF Agent | LLaVA-NeXT-8B | 14.10 | 14.97 | 25.08 |
| HF Agent | InternVL2-8B | 32.05 | 36.75 | 52.18 |
| HF Agent | MiniCPM-V-8.5B | 33.97 | 36.59 | 56.10 |
| HF Agent | Qwen2-VL-7B | 42.31 | 44.85 | 65.19 |
| **T3-Agent** | Tuned MiniCPM-V-8.5B | 52.56 | 65.85 | 80.49 |
| **T3-Agent** | Tuned Qwen2-VL-7B | 53.85 | 64.63 | 84.32 |

Table 3: Results on the validation set of the GAIA benchmark

| Method | Controller | AnsAcc | Level 1 | Level 2 | Level 3 |
|---|---|---|---|---|---|
| Sibyl Agent | GPT-4-turbo | 29.70 | 43.40 | 27.90 | 7.70 |
| Warm-up Act | GPT-4-turbo | 17.60 | 30.20 | 15.10 | 0.00 |
| HF Agent | GPT-4o | 33.40 | 47.17 | 31.40 | 11.54 |
| HF Agent | GPT-4o mini | 26.06 | 33.96 | 27.91 | 3.84 |
| HF Agent | LLaVA-NeXT-8B | 3.64 | 9.43 | 1.16 | 0.00 |
| HF Agent | InternVL2-8B | 4.85 | 7.55 | 4.65 | 0.00 |
| HF Agent | MiniCPM-V-8.5B | 7.27 | 13.21 | 5.81 | 0.00 |
| HF Agent | Qwen2-VL-7B | 9.70 | 16.98 | 8.14 | 0.00 |
| **T3-Agent** | Tuned MiniCPM-V-8.5B | 15.15 | 26.42 | 11.63 | 3.84 |
| **T3-Agent** | Tuned Qwen2-VL-7B | 16.97 | 26.42 | 15.12 | 3.84 |

perception, operation, logic, and creativity abilities on visual data. In addition to visual data, diverse files (such as PPTX, PDF, XLSX files, *etc*) are also commonly encountered in practical multi-modal tasks. To evaluate agents on such files, we use the GAIA benchmark that contains 446 tasks with 109 files. The tasks in GAIA are divided into three levels, the steps of which range from 2 to arbitrarily long sequences, evaluating the capabilities of document understanding, web surfing, logic reasoning, and answer summarization.

**Metric.** In the GTA benchmark, we measure three metrics for agents, including *AnsAcc*, *ToolAcc*, and *CodeExec*. *AnsAcc* measures the correctness of predicted answers. *ToolAcc* means the accuracy of tool selection and answer summary. *CodeExec* quantifies the percentage of generated codes that could be executed without errors. In the GAIA benchmark, we measure *AnsAcc* of its three levels.

## 5.2 GTA RESULTS

The performance of agents on the GTA benchmark is shown in Tab. 2, where *AnsAcc*, *ToolAcc*, and *CodeExec* are reported. Our agent achieves better results than the Lego agent that uses closed-source models (*e.g.*, GPT-4 and GPT-4o) and HF agent using open-source models (*e.g.*, InternVL2-8B), showing its effectiveness in solving complex tasks. The comparison of agents using the tuned and untuned VLMs shows the effectiveness of our multi-modal agent tuning method. For example, tuning MiniCPM-V-8.5B leads to about $18\%$, $29\%$, and $24\%$ improvements in the answer accuracy, tool correctness, and code executability, respectively. In addition, compared to the HF agent using GPT-4o and GPT-4o mini, our agent has higher *ToolAcc* while lower *CodeExec*, showing that our tuned VLM has more powerful reasoning capability for tool usage, while the weak programming capability results in worse *AnsAcc*. This inspires us to develop VLMs for writing codes.

## 5.3 GAIA RESULTS

In Tab. 3, we report the performance of T3-Agent on the validation set of GAIA. T3-Agent performs better than agents driven by open-source models. For example, Qwen2-VL-7B achieves the best performance among all open-source models, while our agent is still $10\%$ higher than it. The performance improvements across multiple VLMs validate the effectiveness of our dataset. Compared with agents driven by closed-source models (*e.g.*, GPT-4), our T3-Agent achieves worse performance.

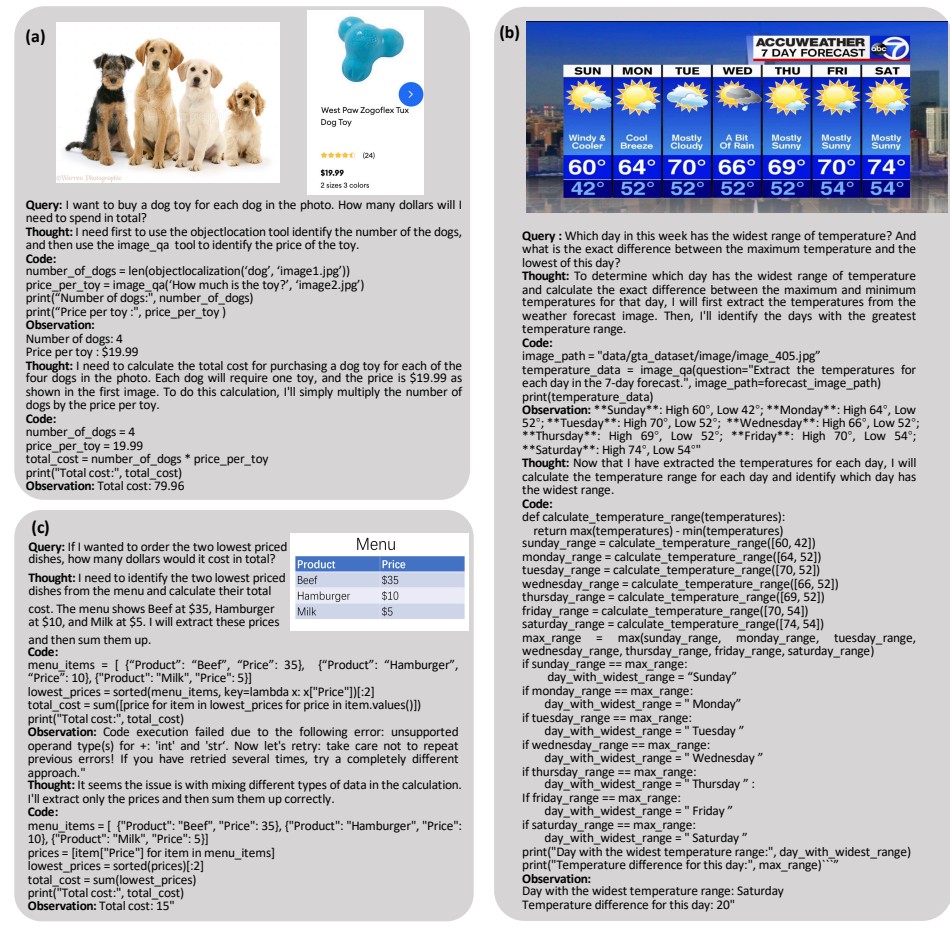

Figure 4: Case study of the T3-Agent in the GTA benchmark.

The reason is that closed-source models use larger model sizes and more training data. These factors may primarily contribute to the performance differences.

## 5.4 DATA QUALITY

To evaluate the data quality of generated data in MM-Traj, we conduct a user study. Concretely, we randomly sample 600 data points from the MM-Traj datasets and filtered out data. We ask 30 persons (with rich programming experience) to provide scores (1-10) for the task quality and trajectory quality, and they do not know whether the data is from MM-Traj or filtered out data. We ask the persons to provide scores for the tasks (the queries and files) and trajectories. The score is in the range (1-10), where higher scores mean better quality. Results are shown in Tab. 4. The quality of MM-Traj is higher than the filtered-out data, demonstrating the used verifiers can discard lower-quality data.

Table 4: Average scores from humans.

|  | MM-Traj | | Filtered out Data | |
| --- | --- | --- | --- | --- |
| Task | Trajectory | | Task | Trajectory |
| 8.32 | 8.67 | | 6.36 | 6.38 |

## 5.5 ABLATION

We conduct ablation experiments to evaluate the effectiveness of our two verifiers, as shown in Tab. 5. We observe that on both the two benchmarks, the data using the two verifiers leads to better performance (*e.g.*, 2.56% improvements on the GTA benchmark), showing the effectiveness of the two verifiers.

Table 5: Ablation

| Method | GTA | GAIA |
| --- | --- | --- |
| w/o two verifier | 50.00 | 13.33 |
| Ours (two verifiers) | 52.56 | 15.15 |

## 5.6 VISUALIZATION

In Fig. 4 and Fig. 5, we visualize cases solved by our T3-Agent in the GTA and GAIA benchmarks. We have the following conclusions. (1) Our agent could handle multiple-image reasoning tasks. By utilizing the visual information from given images, the agent could apply the correct tools and write correct arguments for given images. (2) Our agent could solve complex tasks requiring long code. (3) Our agent could revise code errors based on observations. (4) T3-Agent can solve tasks with multi-hop questions. For example, in the first case of Fig. 5, our agent searches for information from the web, based on obtained information in the first step. (5) T3-Agent could handle multi-modal files, such as audio and PDF files in Fig. 5.

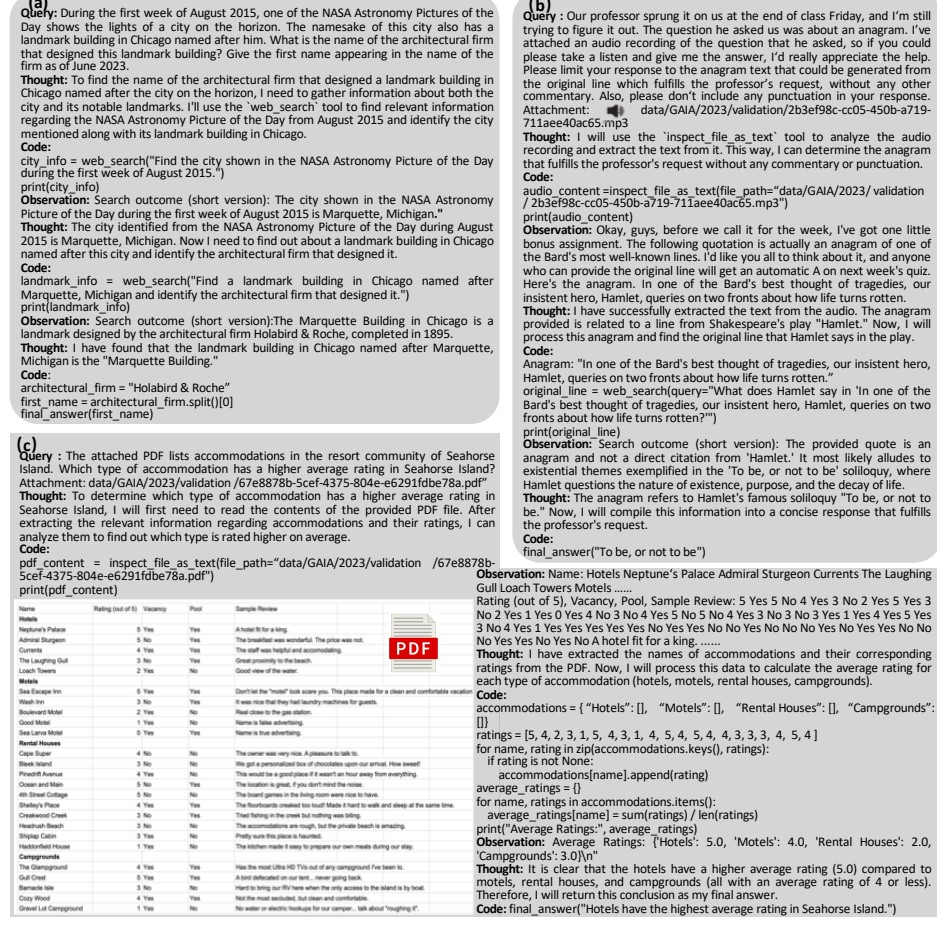

Figure 5: Case study of the T3-Agent in the GAIA benchmark.

## 6 CONCLUSION

In this paper, we have presented a multi-modal agent tuning method that improves the tool-usage capability of agents by generating a large number of tool-usage data and tuning a VLM using these data. Given proper prompts and sufficient source images, our data synthesis pipeline can produce high-quantity multi-modal tasks with trajectories. We collect these generated data into an MM-Traj dataset to tune a MiniCPM-V model, and the T3-Agent with the tuned model has achieved significant improvements on two multi-modal benchmarks, demonstrating the effectiveness of the data synthesis pipeline and the collected MM-Traj dataset.

**Limitation.** In the current T3-Agent, we only consider the multi-modal information in queries. Practical tasks usually involve multi-modal data in trajectories of agents, such as the intermediate results in image editing tasks. We will study how to utilize multi-modal information in agent trajectories, which benefits in performing more powerful step-by-step reasoning for tool usage.

**Acknowledgements.** This work was supported by the China Postdoctoral Science Foundation (No. 2024M760114), National Science and Technology Major Project (2022ZD0114900), China Postdoctoral Science Foundation (No. 2024M760114), the Opening Project of the State Key Laboratory of General Artificial Intelligence (SKLAGI2024OP01, SKLAGI2024OP14), the Natural Science Foundation of China (NSFC) under Grants No. 62172041 and No. 62406009, the Key Program of the National Natural Science Foundation of Shenzhen under Grant No. 202412023000734, and the Key Research Platforms and Projects of the Guangdong Provincial Department of Education under Grant No.2023ZDZX1034.

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

## A HUMAN VERIFICATION OF MM-TRAJ

### A.1 DATA SYSTHESIS PIPELINE

We recruited 30 persons with rich programming and AI experience to evaluate the tasks and trajectories generated by our method. Each evaluator is tasked with assessing 20 samples, which are randomly selected and mixed from both MM-Traj and filtered-out data.

The evaluation was conducted using a 5-level rating scale: **Poor**, **Fair**, **Good**, **Very Good**, and **Excellent**, corresponding to numerical scores of 2, 4, 6, 8, and 10, respectively, with a maximum score of 10. The results in Tab. 6 show that verified cases consistently outperform filtered-out cases in both task and trajectory evaluations. Verified tasks scored an average of 7.96, while filtered-out tasks averaged 6.30, indicating that verified tasks are more natural, coherent, and complex. For trajectories, verified cases scored 8.64 versus 6.24 for filtered-out cases, demonstrating better reasoning, code coherence, and feedback effectiveness. These results confirm that the verification process effectively filters out lower-quality data, ensuring the reliability of our data synthesis pipeline and the MM-Traj dataset.

| MM-Traj | | Filtered out Data | |
| --- | --- | --- | --- |
| Task | Trajectory | Task | Trajectory |
| 8.32 | 8.67 | 6.36 | 6.38 |

Table 6: Average score of the human verification.

The evaluation criteria for the generated tasks and trajectories are as follows.

**Task Evaluation Criteria**

- **Naturalness**: The degree to which the task appears natural and realistic.
- **Coherence**: The logical consistency and smooth flow of the task.
- **Complexity**: The extent to which the task exhibits sufficient complexity, requiring the use of multiple tools for effective resolution.

**Trajectory Evaluation Criteria**

- **Reasoning**: The logical soundness and clarity of the agent's thought process.
- **Code Coherence**: The clarity, consistency, and structure of the code produced by the agent.
- **Feedback Effectiveness**: The agent's ability to effectively respond to and incorporate results from the tool executions.

The interface of the user study for data quality is shown in Fig. 6.

### A.2 AGENT OUTPUT

We conducted a user study on agent outputs on the GTA benchmark. We recruited 20 participants, each evaluating 20 tasks with agent outputs, where the agent is with or without fine-tuning. The agent outputs (w/ or w/o tuning) were shuffled for each task, and the participants were not informed about the source, ensuring an unbiased assessment. The participants were asked to provide the preference of the two agent outputs, based on the accuracy, helpfulness, and relevance. We measured the percentages of results, as shown in Tab. 7. Outputs from the tuned agent have a significantly high preference, indicating its better performance in solving practical tasks. The interface of the user study for agent output is shown in Fig. 7.

| Agent w/o tuning is better | Tie | Agent w tuning is better |
| --- | --- | --- |
| 21% | 13% | 66% |

Table 7: User study for agent outputs on the GTA benchmark.

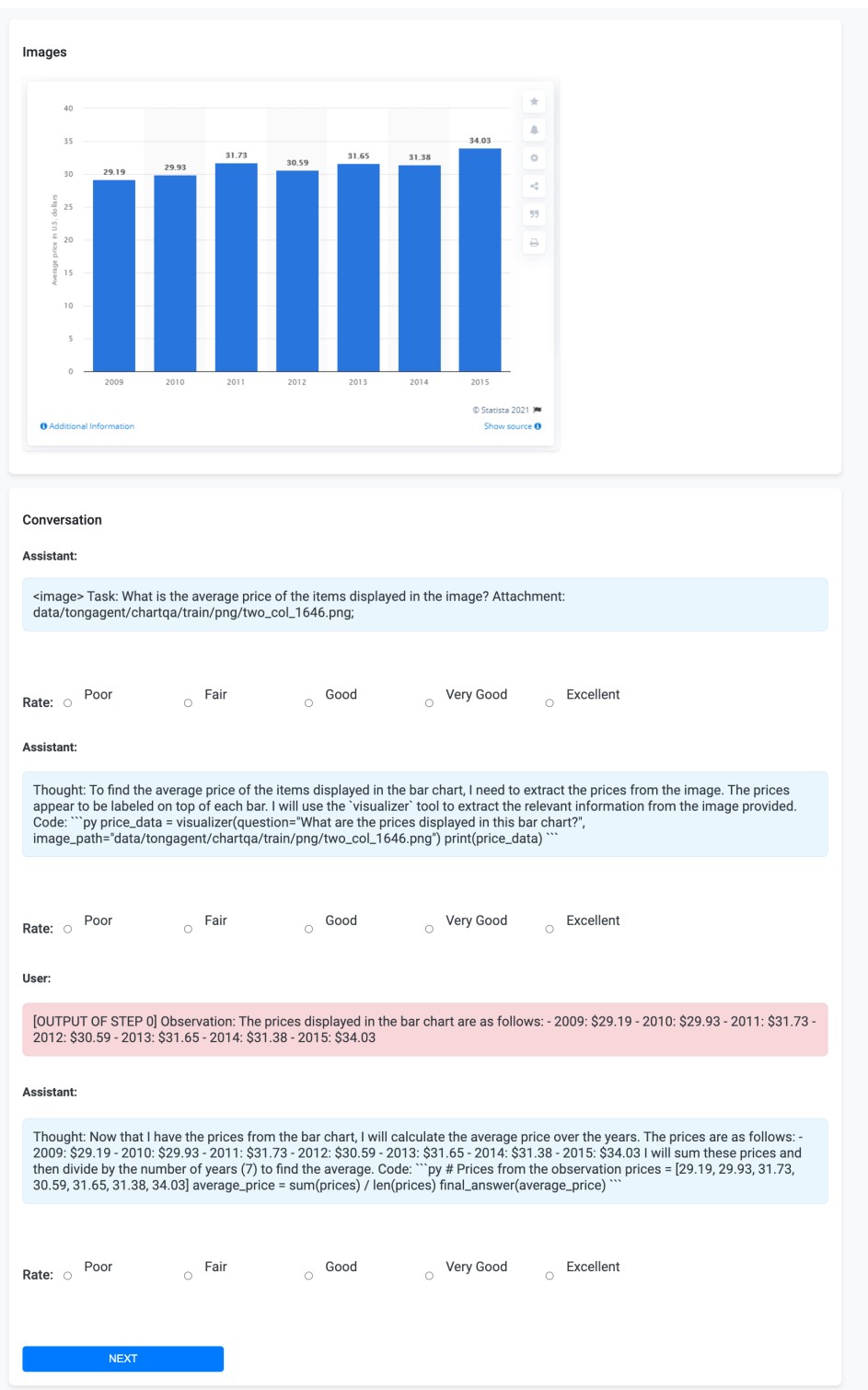

Figure 6: The interface of the user study for data quality.

Please evaluate which model you prefer based on the trajectory and model response.

**Question 9 of 10**

Menu

| Product | Price |
|---------|-------|
| Beef | $10 |
| Hamburger | $20 |
| Juice | $8 |
| Pizza | $36 |

**Task:** Task: If I wanted to order the two highest priced dishes, how many dollars would it cost in total?

**Agent A Trajectory:**

```
[
  {
    "role": "assistant",
    "content": "Thought: I need to find the two highest priced dishes from the menu and calculate
their total cost.\nCode:\n```py\n# Menu data extracted from the image\nmenu_items = [\n    {'Product':
'Beef', 'Price': 10},\n    {'Product': 'Hamburger', 'Price': 20},\n    {'Product': 'Juice', 'Price':
8},\n    {'Product': 'Pizza', 'Price': 36}\n]\n\n# Sort menu items by price in descending
order\nsorted_menu = sorted(menu_items, key=lambda x: x['Price'], reverse=True)\n\n# Get the top two
most expensive items\ntop_two_items = [item['Product'] for item in sorted_menu[:2]]\n\n# Calculate the
total cost of these items\ntotal_cost = sum(item['Price'] for item in
top_two_items)\nprint(total_cost)\n```"
  },
  {
    "role": "tool-response",
    "content": "[OUTPUT OF STEP 0] Error: Code execution failed due to the following error:\n'<='
not supported between instances of 'int' and 'str'\nNow let's retry: take care not to repeat previous
errors! If you have retried several times, try a completely different approach.\n"
  },
  {
    "role": "assistant",
    "content": "Thought: Since the previous code execution failed, I will try a different approach
to find the two highest priced dishes and calculate their total cost.\nCode:\n```py\n# Menu data
extracted from the image\nmenu_items = [\n    {'Product': 'Beef', 'Price': 10},\n    {'Product':
'Hamburger', 'Price': 20},\n    {'Product': 'Juice', 'Price': 8},\n    {'Product': 'Pizza', 'Price':
36}\n]\n\n# Sort menu items by price in descending order\nsorted_menu = sorted(menu_items, key=lambda
x: x['Price'], reverse=True)\n\n# Get the top two most expensive items\ntop_two_items =
[item['Product'] for item in sorted_menu[:2]]\n\n# Calculate the total cost of these items\ntotal_cost
= sum(item['Price'] for item in top_two_items)\nprint(total_cost)\n```"
  },
  {
    "role": "tool-response",
    "content": "[OUTPUT OF STEP 1] Error: Code execution failed due to the following error:\n'<='
not supported between instances of 'int' and 'str'\nNow let's retry: take care not to repeat previous
errors! If you have retried several times, try a completely different approach.\n"
  },
  {
    "role": "assistant",
    "content": "I apologize for the error. Let's try a different approach to find the two highest
priced dishes and calculate their total cost.\nCode:\n```py\n# Menu data extracted from the
image\nmenu_items = [\n    {'Product': 'Beef', 'Price': 10},\n    {'Product': 'Hamburger', 'Price':
20},\n    {'Product': 'Juice', 'Price': 8},\n    {'Product': 'Pizza', 'Price': 36}\n]\n\n# Sort menu
items by price in descending order\nsorted_menu = sorted(menu_items, key=lambda x: x['Price'],
reverse=True)\n\n# Get the top two most expensive items\ntop_two_items = [item['Product'] for item in
sorted_menu[:2]]\n\n# Calculate the total cost of these items\ntotal_cost = sum(item['Price'] for item
in top_two_items)\nprint(total_cost)\n```"
  },
  {
    "role": "tool-response",
    "content": "[OUTPUT OF STEP 2] Error: Code execution failed due to the following error:\n'<='
not supported between instances of 'int' and 'str'\nNow let's retry: take care not to repeat previous
errors! If you have retried several times, try a completely different approach.\n"
  },
  {
    "role": "assistant",
    "content": "I apologize for the error. Let's try a different approach to find the two highest
priced dishes and calculate their total cost.\nCode:\n```py\n# Menu data extracted from the
image\nmenu_items = [\n    {'Product': 'Beef', 'Price': 10},\n    {'Product': 'Hamburger', 'Price':
20},\n    {'Product': 'Juice', 'Price': 8},\n    {'Product': 'Pizza', 'Price': 36}\n]\n\n# Sort menu
items by price in descending order\nsorted_menu = sorted(menu_items, key=lambda x: x['Price'],
reverse=True)\n\n# Get the top two most expensive items\ntop_two_items = [item['Product'] for item in
sorted_menu[:2]]\n\n# Calculate the total cost of these items\ntotal_cost = sum(item['Price'] for item
in top_two_items)\nprint(total_cost)\n```"
  },
  {
    "role": "tool-response",
    "content": "[OUTPUT OF STEP 3] Error: Code execution failed due to the following error:\n'<='
not supported between instances of 'int' and 'str'\nNow let's retry: take care not to repeat previous
errors! If you have retried several times, try a completely different approach.\n"
  },
  {
    "role": "assistant",
    "content": "I apologize for the error. Let's try a different approach to find the two highest
priced dishes and calculate their total cost.\nCode:\n```py\n# Menu data extracted from the
image\nmenu_items = [\n    {'Product': 'Beef', 'Price': 10},\n    {'Product': 'Hamburger', 'Price':
20},\n    {'Product': 'Juice', 'Price': 8},\n    {'Product': 'Pizza', 'Price': 36}\n]\n\n# Sort menu
items by price in descending order\nsorted_menu = sorted(menu_items, key=lambda x: x['Price'],
reverse=True)\n\n# Get the top two most expensive items\ntop_two_items = [item['Product'] for item in
sorted_menu[:2]]\n\n# Calculate the total cost of these items\ntotal_cost = sum(item['Price'] for item
in top_two_items)\nprint(total_cost)\n```"
  },
  {
    "role": "tool-response",
    "content": "[OUTPUT OF STEP 4] Error: Code execution failed due to the following error:\n'<='
not supported between instances of 'int' and 'str'\nNow let's retry: take care not to repeat previous
errors! If you have retried several times, try a completely different approach.\n"
  },
  {
    "role": "tool-response",
    "content": "[OUTPUT OF STEP 5] Error: Reached max execution exception. Max exception
tolerance: 5.\nNow let's retry: take care not to repeat previous errors! If you have retried several
times, try a completely different approach.\n"
  }
]
```

**Agent B Trajectory:**

```
[
  {
    "role": "assistant",
    "content": "Thought: To determine the total cost of ordering the two highest priced dishes, I
will first extract the prices from the menu. Then, I'll identify the two most expensive items and
calculate their combined price.\n\nCode:\n```py\n# Extracting prices from the menu
image\nmenu_image_path = \"/image/image_313.jpg\"\nmenu_prices = visualizer(question=\"What are the
prices in the menu?\", image_path=menu_image_path)\nprint(menu_prices)\n```"
  },
  {
    "role": "tool-response",
    "content": "[OUTPUT OF STEP 0] -> Observation:\nThe prices on the menu are as follows:\n\n-
Beef: $10\n- Hamburger: $20\n- Juice: $8\n- Pizza: $36\n"
  },
  {
    "role": "assistant",
    "content": "Thought: Now that I have the prices, I will identify the two most expensive dishes
and calculate their total cost.\n\nCode:\n```py\n# Prices extracted from the menu\nprices = {\n
\"Beef\": 10,\n    \"Hamburger\": 20,\n    \"Juice\": 8,\n    \"Pizza\": 36\n}\n\n# Identifying the
two highest priced items\nhighest_prices = sorted(prices.items(), key=lambda x: x[1], reverse=True)
[:2]\ntotal_cost = sum([price[1] for price in highest_prices])\nprint(\"Total cost of ordering the two
highest priced dishes:\", total_cost)\n```"
  },
  {
    "role": "tool-response",
    "content": "[OUTPUT OF STEP 1] -> Observation:\nTotal cost of ordering the two highest priced
dishes: 56\n"
  },
  {
    "role": "assistant",
    "content": "Thought: The total cost of ordering the two highest priced dishes is $56. Now, I
will provide this as the final answer.\n\nCode:\n```py\nfinal_answer(\"The total cost to order the two
highest priced dishes is $56.\")\n```"
  },
  {
    "role": "tool-response",
    "content": "[OUTPUT OF STEP 2] -> Observation:\nNo observation found from the code execution.
You should use `print` function if need some information from the code execution."
  }
]
```

**Agent A Final Answer:** I apologize for the repeated errors. Let's try a different approach to find the two highest priced dishes and calculate their total cost. Code: ```py # Menu data extracted from the image menu_items = [ {'Product': 'Beef', 'Price': 10}, {'Product': 'Hamburger', 'Price': 20}, {'Product': 'Juice', 'Price': 8}, {'Product': 'Pizza', 'Price': 36} ] # Sort menu items by price in descending order sorted_menu = sorted(menu_items, key=lambda x: x['Price'], reverse=True) # Get the top two most expensive items top_two_items = [item['Product'] for item in sorted_menu[:2]] # Calculate the total cost of these items total_cost = sum(item['Price'] for item in top_two_items) print(total_cost) ```

**Agent B Final Answer:** The total cost to order the two highest priced dishes is $56.

**Task:** Task: If I wanted to order the two highest priced dishes, how many dollars would it cost in total?

**Groundtruth:** [['56']]

| 👉 A is Better | 👈 Tie | 👉 B is Better |

Figure 7: The interface of the user study for agent outputs on the GTA benchmark.

## B    MORE EXPERIMENTS

### B.1    ABLATION

To improve the interpretability of our method, we add ablation experiments to show the contributions of different modalities in decision-making. Ablation results on the GTA dataset are shown in Tab. 8, where removing the image modality reduces the performance by $40\%$, highlighting the importance of input images.

| Method | AnsAcc | ToolAcc | CodeExec |
|---|---|---|---|
| T3 Agent w/o image | 10.67 | 25.32 | 20.09 |
| T3 Agent w/ image | **52.56** | **65.85** | **80.49** |

Table 8: Average score of the human verification.

### B.2    DATA NUMBER

We show the agent's performance on the GTA benchmark as the dataset size increases, in Tab. 9. With the increase of data number, the agent achieves better performance, the memory consumption is constant, and the time consumption linear increases. Compared with the accuracy improvements, we think that the consumption of memory and time is acceptable.

| Data Number | 6K | 12K | 20K |
|---|---|---|---|
| Accuracy | 43.59% | 48.08% | 52.56% |
| Memory | 214 GB | 214 GB | 214 GB |
| Training Time | 276 mins | 532 mins | 946 mins |

Table 9: Performance on the GTA benchmarks.

## C    TOOLS

We will show details of used tools in T3-Agent.

### C.1    WEB SEARCH

The web search tool is actually another agent. It has three sub-tools: *Searchinformation*, *Visit*, and *Webqa*.

*searchinformation*. Given a query to this tool, it performs a Google search and outputs the title, abstract, and URL of multiple entries.

*Visit*. The input is the URL of an HTML page, and the output is the textual content of the HTML page.

*Webqa*. Given a question and search textual content, the tool outputs the answer.

### C.2    IMAGE QUESTION ANSWERING

We use the GPT-4o-mini model as the image question answering tool. The input is an image and a question, and the output is the answer.

### C.3    FILE INSPECTOR

The input is a question and one multi-modal file. We use the Python package 'MarkdownConverter' that converts the given files into the markdown texts. Then, we feed the question and texts to the GPT-4o-mini model for the answer.

### C.4 OBJECT LOCALIZATION

We use the OWL-ViT model (Minderer et al., 2022) for object localization. The input includes one image and a query, and the output is a Python list of bounding boxes of the query.

### C.5 IMAGE GENERATION

We use the stable diffusion model for image generation (Rombach et al., 2022). Given a textual query, the tool produces an image that matches the query.

### C.6 IMAGE EDITING

We use the InstructPix2Pix model for image editing (Brooks et al., 2023). The inputs are an instruction and an image, and the output is an edited image to match the instruction.

### C.7 FACE DETECTION

We use the DSFD model for face detection (Li et al., 2019). The input is an image, and the output is a Python list of all bounding boxes of faces.

### C.8 PYTHON PACKAGE

We allow the agent to use the following Python package in code writing: "requests", "zipfile", "os", "pandas", "numpy", "sympy", "json", "bs4", "pubchempy", "xml", "yahoo_finance", "Bio", "sklearn", "scipy", "pydub", "io", "PIL", "chess", "PyPDF2", "pptx", "torch", "datetime", "csv", "fractions", "matplotlib", "pickle", "cv2", through which the agent is more flexible in writing code.

## D PROMPT

### D.1 PROMPT FOR QUERY GENERATION

The prompt for query generation is shown in Fig. 8.

> You are tasked with generating user queries that will prompt an agent to call various tools (only use the tool listed in our toolset), including internet search capabilities, to solve real-world, practical problems. The problems should be natural, varied, and challenging, requiring the agent to reason across different domains. Ensure that the problems span a range of practical scenarios.
>
> Our toolset: TOOL_SET
> I will now provide examples, along with the tools.
> Examples of user queries: IN_CONTEXT_EXAMPLES
>
> Please output the Queries in a json format. Make sure that the queries share a similar style of the in-context examples. The output template is :
> {
>  "query": "What is the weather today?", <The user query to the agent.>
>  "tools": ["tool1", "tool2",...], <A list consisting of the tool names related to the query.>
> },
> ...

Figure 8: Prompt for query generation.

## D.2 PROMPT FOR FILE GENERATION

The prompt for file content generation is shown in Fig. 9 and Fig. 10 , and the prompt for file code generation is shown in Fig. 11 and Fig. 12.

---

You are a smart reasoner that can restore a query_solving scene between a human and an agent. Human gives a complex query and several images to the agent, and then the agent answers the query by searching on the Internet and applying tools to the images with step-by-step reasoning. Now, you will be given the query with suggested tools, I suggest you analyze the needed information to solve the query, and divide the information into two groups: searching from the Internet and extracting from the images using tools. Based on the information from the images, you need to further infer the content of these images, through which the agent could correctly solve the query.

Our toolset: TOOL_SET
Output MUST use the following json template.

```
{
    "information": <Needed information to solve the query. For the query including creating/generating images, the information should NOT be the description of the describe image.>
    "information from the Internet": <Information from the Internet inferences based on the given query and suggested tools. Determine which information is suitable to be obtained from the Internet. Or say no information is required from the Internet.>
    "information from images": <Information extracted from given images based on the suggested tools to solve the query. It should be several sentences, including information extracted from the images using tools. Determine which information is suitable to be obtained from the images, and using which tools. Do not generate image_content for the query including generating/creating an image. Or say no information is required from the images.>
    "file": {
        "image_numbers": <set an int number, the number is depended on needed information from images>,
        "image_content":
        {
            "image_1": <The image content should be a natural language, describe the content of the first image relevant to the query. The content should be concrete, such as concrete number, concrete name. The content should match the query and the above images.>
            ... <if you think the query needs more than 1 image, please output image content like 'image_2'.>
        }
    }
}
```

Figure 9: System prompt for the file content generation.

---

Now given the query: QUERY, firstly analyze the needed information to solve the query and divide the information into two groups: searching from the Internet or extracting from images using tools. Then for information from images, imagine possible answers of each information (it should be concrete answers instead of descriptions). Finally, output the json for the inferenced information and the content of images.

Figure 10: User prompt for the file content generation.

---

## D.3 PROMPT FOR QUERY-FILE FILTER

The prompt for the query-file filter is shown in Fig. 13 and Fig. 14.

You are a helpful assistant and can to generate a <file type placeholder> file by writing Python code. You will be given a description of the content of the file. You need to firstly largely extend the content, and then write Python code to generate a <file type placeholder> file. GUARANTEE that the provided content is in the file.
The output Python code MUST use the following template.
"""

    ## extention start
        Extened content: <here is the extented content>
    ## extention end

    ## code start
        <here is the Python code to generate a <file type placeholder> file>
    ## code end
"""

Figure 11: User prompt for the file content generation.

Now, given the following content: <file content>, first largely extend the content, and output a code to generate a <file type placeholder> file, where the file name is <file name> and the file will be saved in <save path>.

Figure 12: User prompt for the file content generation.

## D.4 PROMPT FOR TRAJECTORY FILTER

The prompt for the trajectory filter is shown in Fig. 15 and Fig. 16.

## D.5 PROMPT FOR AGENTS

The system prompt for the T3-Agent is shown in Fig. 17.

# E MORE VISUALIZATION

We provide more visualization of our T3-Agent on the GTA and GAIA benchmarks, as shown in Figs. 18 to 24.

You are a helpful assistant that is given a query and several images. You need to check whether the images are relevant to the query. The query and images are used to evaluate the perception ability, reasoning ability, and information search ability of an AI agent. The agent solves the query by searching information from the Web and extracting information from the images. In some cases, based on the given images, the agent could not solve the query, even it searches for information from the Web (e.g., some specific knowledge). You need to pick up these bad cases.

The agent can call the following tools to solve the query. TOOL_SET .

Thus, the images should follow these requirements.
1. Relevance: The depicted scenarios or objects in images should be relevant to the query. The images should contain scenarios or objects that are mentioned in the images.
2. Usefulness: The image should contain necessary information to address the query, such as some specific details that cannot be obtained from the Web.
3. Some queries require the agent to search for knowledge from the Web and combine the information in the image to solve the queries. Thus, in some cases, the images do not contain all the information to solve the query, but the missed information could be searched on the Web. These cases should be regarded as correct cases.

The output MUST use the following json template to check the images.
{     "information_for_query": <Required information to solve the query.>,
    "useful_information_in_image": <Useful information that can be extracted from images to solve the query>,
    "missed_information_in_images": <Missed information that is necessary to solve the query but does not exist in the images.>,
    "missed_information_web_search": <You need to justify whether the missed information could be searched from the Web, using your rich experience in surfing the Internet.> ,
    "missed_information_obtained": <You need to justify whether the missed information could be obtained via computing or reasoning based on information extracted from the images or searched from the Web.>,
    "thought": <Now, you need to determine whether the images can solve the query. If the missed information could be searched from the Web or obtained based on existing information, the images can solve the query. If not, the images cannot solve the query.>,
    "correct": <According to the above reasoning, if you consider the images reasonable for the query to be solved by the tools, set the value to 'yes', otherwise set the value to 'no'.>,
    "updated_query": <If you judge the correctness as 'no', please rewrite the query to make it more relevant to the given images. If you judge the correctness as 'yes', please output "no revision is needed." >
} ”'

Figure 13: System prompt for the query-file verification.

Following are images, the query: <query>, inference whether the images can solve the query based on the perception ability, reasoning ability, and information search ability of an AI agent.

Figure 14: User prompt for the query-file verification.

As a data quality evaluator that need to determine whether a query-solving trajectory between a human and an agent is correct. The human gives images and a query, and the agent calls tools to solve the query. The trajectory of query-solving contains a task query, thoughts and codes generated by the agent to call tools (Python functions), and tool-response of each step, and the final answer. You must assess the alignment between the task query, corresponding tool usage (generated thoughts and codes from the agent), and the execution results (tool-response). Your goal is to ensure the used tools, arguments to the tools, and summarized answers in the trajectory accurately reflect the human's intentions.
Our toolset: TOOL_SET
The query-solving trajectory is incorrect if: 1. The tool usage does not align with the query's objective and the context, or there is useless or unreasonable tool usage. In addition, the agent does not use tools and solve the query by itself. 2. The input arguments to the tools appear incorrect or unreasonable. 3. The final answers or intermediate results summarized from the observation appear incorrect or unreasonable. 4. The final answer is not relevant to the task query or the final answer seems incorrect. 5. The trajectory (such as tool-usage and observation) conflicts or is not consistent with the image content.

Figure 15: System prompt for the trajectory verification.

Now, given used images and corresponding information, determine whether the trajectory is correct or not.

1. User Query: QUERY
2. Image Content: IMAGE_CONTENT
3. Trajectory, including generated thought and code from the agent, and intermediate results of using tools: TRAJ
4. Execution Results: RESULT
Output MUST use the following json template to determine whether the query-solving trajectory is correct or not.
{
"thought": "Concisely describe your reasoning here",
"correct": "yes" or "no"
}

Figure 16: User prompt for the trajectory verification.

You are an expert assistant who can solve any task using code blobs. You will be given a task to solve as best you can. To do so, you have been given access to a list of tools: these tools are basically Python functions which you can call with code. To solve the task, you must plan forward to proceed in a series of steps, in a cycle of 'Thought:', 'Code:', and 'Observation:' sequences.

At each step, in the 'Thought:' sequence, you should first explain your reasoning towards solving the task and the tools that you want to use. Then in the 'Code:' sequence, you should write the code in simple Python. The code sequence must end with the '<end_action>' sequence. During each intermediate step, you can use 'print()' to save whatever important information you will then need. DO NOT generate a code which does not call 'print()' because you will lose this information. You can assume all tools must have a return that can be printed. These print outputs will then appear in the 'Observation:' field, which will be available as input for the next step. You will save all intermediate file outputs to a folder by the relative path '.cache'. In the end, you have to return a final answer using the 'final_answer' tool.
Here are a few examples using notional tools: IN_CONTEXT_EXAMPLES

The above example was using notional tools that might not exist for you. You only have access to those tools: TOOL_SET
You also can perform computations in the Python code that you generate.

Here are the rules you should always follow to solve your task:
1. Always provide a 'Thought:' sequence, and a 'Code: py' sequence ending with '<end_action>' sequence, else you will fail.
2. Use only variables that you have defined!
3. Always use the right arguments for the tools. DO NOT pass the arguments as a dict as in 'answer = ask_search_agent('query': "What is the place where James Bond lives?")', but use the arguments directly as in 'answer = ask_search_agent(query="What is the place where James Bond lives?")'.
4. Take care to not chain too many sequential tool calls in the same code block, especially when the output format is unpredictable. For instance, a call to search has an unpredictable return format, so do not have another tool call that depends on its output in the same block: rather output results with print() to use them in the next block.
5. Call a tool only when needed, and never re-do a tool call that you previously did with the exact same parameters.
6. Don't name any new variable with the same name as a tool: for instance don't name a variable 'final_answer'.
7. Never create any notional variables in our code, as having these in your logs might derail you from the true variables.
8. You can use imports in your code, but only from the following list of modules: «authorized_imports»
9. The state persists between code executions: so if in one step you've created variables or imported modules, these will all persist.
10. Don't give up! You're in charge of solving the task, not providing directions to solve it.

Now Begin! If you solve the task correctly, you will receive a reward of $1,000,000.

Figure 17: System prompt for the agent.

| RANK | COMPANY |
|------|---------|
| 1 | APPLE |
| 2 | TESLA |
| 3 | amazon |
| 4 | Alphabet |
| 5 | Microsoft |

**Query :** Who is the CEO of the top-ranked company?
**Thought:** To find the CEO of the top-ranked company, I need to analyze the provided image. The task involves identifying which company is ranked first and then determining its CEO. I will use the `image_qa` tool to extract any textual information from the image.
**Code:**
```
image_path = "data/gta_dataset/image/image_399.jpg"
company_rank = image_qa(question="What is the name of the top-ranked company?", image_path=image_path)
print("Top-ranked company:", company_rank)
```
**Observation:** Top-ranked company: The top-ranked company is Apple.
**Thought:** Now that I know the top-ranked company is Apple, I need to find out who its CEO is. Since this information may not be directly visible in the image, I will use the `ask_search_agent` tool to search for the current CEO of Apple.
**Code:**
```
ceo_of_apple = ask_search_agent(query="Who is the CEO of Apple?")
print("CEO of Apple:", CEO_of_Apple)
```
**Observation:** Search outcome (short version): Tim Cook is the current CEO of Apple Inc., having assumed the role in August 2011. Under his leadership, Apple has seen a significant increase in revenue and profit, and a strategic shift towards a more inclusive company culture.
**Thought:**
I have gathered information about Tim Cook being the current CEO of Apple. Now, I will summarize this information to provide a concise final answer.
**Code:**
```
final_answer("Tim Cook is the current CEO of Apple.")
```

Figure 18: Case study of the T3-Agent.

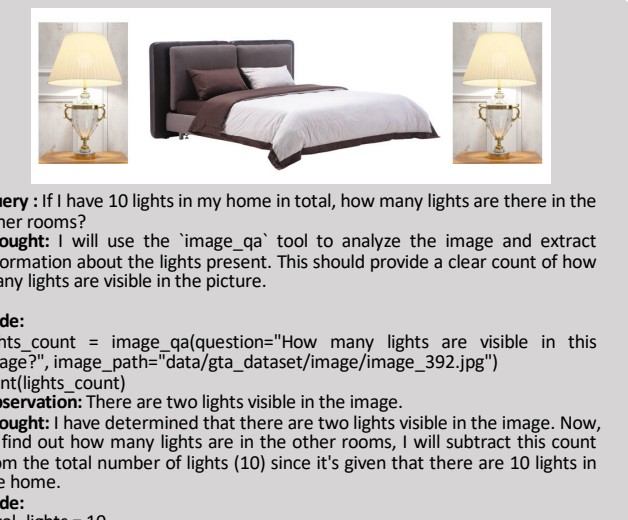

**Query :** If I have 10 lights in my home in total, how many lights are there in the other rooms?
**Thought:** I will use the `image_qa` tool to analyze the image and extract information about the lights present. This should provide a clear count of how many lights are visible in the picture.

**Code:**
```
lights_count = image_qa(question="How many lights are visible in this image?", image_path="data/gta_dataset/image/image_392.jpg")
print(lights_count)
```
**Observation:** There are two lights visible in the image.
**Thought:** I have determined that there are two lights visible in the image. Now, to find out how many lights are in the other rooms, I will subtract this count from the total number of lights (10) since it's given that there are 10 lights in the home.
**Code:**
```
total_lights = 10
lights_in_other_rooms = total_lights – 2
final_answer(lights_in_other_rooms)
```

Figure 19: Case study of the T3-Agent.

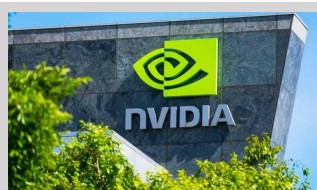

**Query : W**ho is the founder of this company ?
**Thought:** To find out who the founder of the company depicted in the image, I will use the `image_qa` tool to analyze the image and extract relevant information about the company. This should help identify key details such as its name and any associated individuals.
**Code:**
company_info = image_qa(question="What is the name of the company shown in this image?", image_path="image_306.jpg\")
print(company_info)
**Observation:** The company shown in the image is NVIDIA.
**Thought:** Now that I know the company is NVIDIA, I will use the `web_search` tool to find out who founded it. This should provide me with the name of the founder.
**Code:**
founder = web_search(query="Who is the founder of NVIDIA?")
print(founder)
**Observation:** Search outcome (short version): NVIDIA was founded on April 5, 1993, by Jensen Huang, Chris Malachowsky, and Curtis Priem.
**Thought:**
I have gathered information about the founders of NVIDIA. The company was founded by Jensen Huang, Chris Malachowsky, and Curtis Priem. Now, I will compile this information into a final answer.
**Code:**
founders = ["Jensen Huang", "Chris Malachowsky",   "Curtis Priem"]
final_answer(f"Founders of NVIDIA: {', '.join(founders)}")

Figure 20: Case study of the T3-Agent.

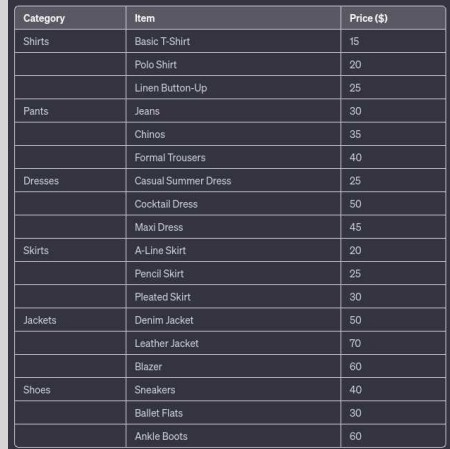

| Category | Item | Price ($) |
|---|---|---|
| Shirts | Basic T-Shirt | 15 |
| | Polo Shirt | 20 |
| | Linen Button-Up | 25 |
| Pants | Jeans | 30 |
| | Chinos | 35 |
| | Formal Trousers | 40 |
| Dresses | Casual Summer Dress | 25 |
| | Cocktail Dress | 50 |
| | Maxi Dress | 45 |
| Skirts | A-Line Skirt | 20 |
| | Pencil Skirt | 25 |
| | Pleated Skirt | 30 |
| Jackets | Denim Jacket | 50 |
| | Leather Jacket | 70 |
| | Blazer | 60 |
| Shoes | Sneakers | 40 |
| | Ballet Flats | 30 |
| | Ankle Boots | 60 |

**Query :** I have only $70 to buy a pants and shoes.I don't wear flats. Each item should be different in style and color. What specific items can I buy? Please answer with a short noun phrase.

**Thought:** I need to identify a pair of pants and shoes that are different in style and color within my budget. Since I have $70, I'll look for options under the given price range. I will use the `image_qa` tool to analyze the image provided and extract relevant information about clothing items, focusing on styles and colors.

**Code:**
```
cloth_items = image_qa(question="Please identify specific pants and shoes from the image based on their styles and colors.", image_path="image_287.jpg")
print(cloth_items)
```

**Observation:** **Pants:**- Jeans ($30) - Chinos ($35)- Formal Trousers ($40) **Shoes:** Sneakers ($40)- Ballet Flats ($30)- Ankle Boots ($60)

**Thought:** I see that the table lists specific clothing items and their prices. However, it doesn't provide visual information about styles or colors of pants and shoes directly from the image provided. Since I don't have a clear visual reference for the specific items, I will summarize the options available based on the listed prices:- **Pants:** Jeans ($30), Chinos ($35), Formal Trousers ($40)- **Shoes:** Sneakers ($40), Ballet Flats ($30), Ankle Boots ($60)Given my budget of $70, I need to choose two different items—one pair of pants and one pair of shoes—without buying flats. I'll propose a combination where I buy one type of jeans and sneakers, ensuring they are distinct in style and color.

**Code:**
```
item1 = "Jeans"
price1 = 30
item2 = "Sneakers"
price2 = 40
total_cost = price1 + price2
print("Total cost:", total_cost)
```

**Observation:** Total cost: 70

**Thought:** I have confirmed that the total cost of buying jeans and sneakers is $70, which matches my budget. Now, I will finalize this purchase.\n\nI'll return the specific items with a short noun phrase to indicate what I can buy within the given constraints

**Code**:
```
final_answer("Jeans and Sneakers")
```

Figure 21: Case study of the T3-Agent.

**Query :** Take the gender split from the 2011 Bulgarian census about those who have completed tertiary education. Subtract the smaller number from the larger number, then return the difference in thousands of women. So if there were 30.1 thousand more men, you'd give 30.1.

**Thought:** I will first need to gather the data regarding the gender split from the 2011 Bulgarian census for those who have completed tertiary education. Once I have these numbers, I can subtract the smaller number from the larger number and return the result in thousands of women.\n\nI'll start by using the `web_search` tool to find the specific data related to the gender split from the 2011 census on those with tertiary education.

**Code:**
gender_data = web_search("gender split from the 2011 Bulgarian census about those who have completed tertiary education")
print(gender_data)

**Observation:** Search outcome: In the 2011 Bulgarian Census, 22.3% of women and 16.7% of men had completed tertiary education.

**Thought:** I have retrieved the relevant data from the 2011 Bulgarian census. According to the information, there were 791.8 thousand females

(representing 22.3% of women) and 556.9 thousand males (representing 16.7% of men) who had completed tertiary education. Now, I will calculate

the difference between these two numbers in thousands of women by subtracting the smaller number from the larger number:- Larger number

(females): 791.8\n- Smaller number (males): 556.9. The difference is calculated as follows: [ text{Difference} = 791.8 - 556.9 ] I'll perform this

calculation using Python code.

**Code:**
larger_number = 791.8
smaller_number = 556.9
difference = larger_number - smaller_number
print("Difference in thousands of women:", difference)

**Observation:** Difference in thousands of women: 234.9

**Thought:** The difference between the numbers of women and men who completed tertiary education in the 2011 Bulgarian census is approximately 234.9 thousand. Now, I will return this result as the final answer using the `final_answer` tool.

**Code:** final_answer("The difference in thousands of women from those with tertiary education in the 2011 Bulgarian census is approximately 234.9.)

Figure 22: Case study of the T3-Agent.

**Query :** The longest-lived vertebrate is named after an island. According to Wikipedia as of January 1, 2021, what is the 2020 estimated population of that island, to the nearest thousand?

**Thought:** To solve this task, I need to find the name of the island after which the longest-lived vertebrate is named. Then, I will search for the estimated population of that island as of January 1, 2020. Since Wikipedia was referenced in the question, I'll use the `web_search` tool to gather this information.

**Code:**
```
longest_lived_island = web_search(query="The name of the island after which the longest-lived vertebrate is named")
print("Island name:", longest_lived_island)
population_query = web_search(query=f"The estimated population of {longest_lived_island} as of January 1, 2020.")
print("Estimated population (to nearest thousand):", population_query)
```

**Observation:** Search outcome: The longest-lived vertebrate is the Greenland shark, named after the island of Greenland.

**Thought:** From the search results, I have found that the longest-lived vertebrate is the Greenland shark. The estimated population of Greenland as of January 1, 2020, was not explicitly mentioned in the provided information. However, since it's related to a specific island, I can assume this refers to the population of Greenland itself. As there is no direct data on the population of Greenland from 2020, I will need to use general knowledge or perform another search for an estimate. I will ask the `web_search` again for an approximate population of Greenland in 2020 and round it to the nearest thousand.

**Code:**
```
population_greenland = web_searc(query="What is the estimated population of Greenland as of January 1, 2020?")
print("Estimated population of Greenland (to nearest thousand):", population_greenland)
```

**Observation:** Search outcome: Greenland's estimated population as of January 1, 2020, was 56,081.

**Thought:** The estimated population of Greenland as of January 1, 2020, is reported to be approximately 56,081. I will now format this information and use the `final_answer` tool to provide a concise answer regarding both the longest-lived vertebrate (Greenland shark) and its associated island's population.

**Code:**
```
population_info = "The longest-lived vertebrate is the Greenland shark, named after the island of Greenland, with an estimated population of approximately 56,000 in 2020."
final_answer(population_info)
```

Figure 23: Case study of the T3-Agent.

**Query :** The Metropolitan Museum of Art has a portrait in its collection with an accession number of 29.100.5. Of the consecrators and co-consecrators of this portrait's subject as a bishop, what is the name of the one who never became pope?

**Thought:** To find the consecrators and co-consecrators of the portrait's subject as a bishop, specifically looking for those who never became pope, I will need to search for information about this particular portrait in the Metropolitan Museum of Art collection. This may involve checking online databases or museum archives related to this specific artwork. \n\nI'll start by asking the `web_search` tool to provide details on the portrait with accession number 29.100.5 at the Metropolitan Museum of Art.

**Code:**
portrait_info = web_search("Details about the portrait with accession number 29.100.5 at the Metropolitan Museum of Art")
print(portrait_info)

**Observation:** Search outcome:- Title: Cardinal Fernando Niño de Guevara n- Artist: El Greco (Domenikos Theotokopoulos) - Date: ca. 1600 - Medium: Oil on canvas - Location: Metropolitan Museum of Art, Accession Number: 29.100.5

**Thought:** The search results indicate that the portrait is of Cardinal Fernando Niño de Guevara, painted by El Greco around 1600. I now need to find out who were the consecrators and co-consecrators of this cardinal subject as a bishop, specifically identifying those who never became pope. I will ask the `web_search` tool again for details on the consecrators and co-consecrators of Cardinal Fernando Niño de Guevara, focusing particularly on those who did not become pope.

**Code:**
consecrators_info = web_search("Consecrators and co-consecrators of Cardinal Fernando Niño de Guevara from the Metropolitan Museum of Art collection, specifically those who never became pope.")
print(consecrators_info)

**Observation:** Search outcome (short version):- Cardinal Fernando Niño de Guevara was consecrated by Pope Clement VIII, with co-consecrators Camillo Borghese (later Pope Paul V) and Alfonso Visconti.- Alfonso Visconti was not a pope.- No direct connection to the Metropolitan Museum of Art was found for Alfonso Visconti.

**Thought:** Based on the search results, I have identified that Cardinal Fernando Niño de Guevara was consecrated by Pope Clement VIII with co-consecrators Camillo Borghese and Alfonso Visconti. Among these individuals, Alfonso Visconti is noted as never having become a pope. I will now compile this information into a final answer format to present it clearly.

**Code:**
final_answer("The co-consecrator of Cardinal Fernando Niño de Guevara who never became pope is Alfonso Visconti.")

Figure 24: Case study of the T3-Agent.

