# OpenReview forum: "Multi-modal Agent Tuning: Building a VLM-Driven Agent for Efficient Tool Usage"
_ICLR.cc/2025/Conference — ICLR 2025 Spotlight_

### Official Review · Reviewer_snR1 · 2024-10-31

**Soundness:** 3
**Presentation:** 3
**Contribution:** 2
**Rating:** 8
**Confidence:** 3

**Summary:**

This work introduces a multi-modal tool-usage data generation pipeline designed to finetune vision-language models (VLMs) for tasks requiring tool-usage reasoning. The pipeline consists of three primary steps. First, a large language model (LLM) is prompted to generate query tasks. Next, relevant images or files are retrieved or generated based on the specified query tasks. Finally, ReAct agents are employed to generate trajectories that address the query task problem, followed by an additional LLM prompt to verify the generated data.

This study also introduces the MM-Traj dataset generated through the proposed scheme and uses it to finetune the MiniCPM-V model to create the T3-agent. The T3-agent's effectiveness is subsequently assessed on the GTA and GAIA benchmarks, showcasing a 20% improvement in performance over untrained VLMs and achieving results comparable to other baseline agents, such as the Lego Agent, Warm-up Act Agent, and HF Agent.

**Strengths:**

1. The proposed data generation pipeline first generates queries independently of any specific files, followed by producing relevant files to align with these queries. This approach allows the pipeline to create more diverse and expressive tasks, unrestricted by file format or quantity limitations.
2. This work introduces the novel MM-Traj dataset, containing 20k diverse tool-usage data points, supported by a human study to demonstrate dataset quality.
3. This work also performs an in-depth statistical analysis of the dataset and shows that MM-Traj offers broad coverage across various data modalities and knowledge requirements, as shown in Figure 3.

**Weaknesses:**

1. In Section 3.3, this work states, “for other needed files, we prompt GPT-4o mini to extend the file content and generate Python code to produce the files.” However, the methodology for file generation remains unclear. For example, if a file is required to be an MP4 video of Taylor Swift’s latest album, it’s uncertain how this content could be generated through Python code alone. Furthermore, if GPT-4o mini generates the Python code to produce such files, it raises concerns about data quality and how the model ensures that the generated content is not hallucinated.
2. While including a human study to assess dataset quality is commendable, having only five experienced raters for a subset of the data may be too limited, potentially introducing biases based on individual preferences. Gathering feedback from a **larger pool of participants**, even with fewer data points per person, could strengthen claims about the dataset's effectiveness. Additionally, comparing MM-Traj to filtered-out data may not yield meaningful insight. Instead, comparisons with other established tool-usage datasets would likely provide more meaningful insights.
3. The evaluation results in Table 3 reveal mixed outcomes, with the T3-agent performing significantly worse than other methods, such as HF Agent and Sibyl Agent, on the GAIA benchmark. What could lead to this performance discrepancy on the GAIA benchmark?

**Questions:**

1. In Figure 3, the sum of trajectories—214 + 14,273 + 8,740 + 2,520 + 1,242 + 697 + 199 + 202—totals 28,087, which exceeds the stated 20k tasks in the abstract. In addition, the paper mentions that only 15k files remain after passing the query-file and trajectory verifiers. What is the final size of the generated dataset?.
2. Why does GPT-4o mini outperform GPT-4o in Table 2, specifically in the row with HF Agents on AnsACC and CodeExec, given that GPT-4o is expected to be more powerful than GPT-4o mini?

---

> ### Author Response · Authors · 2024-11-20
> **Response To Reviewer snR1 [1/2]**
>
> Thanks for your insightful and thorough review. We will address your concerns below.
>
> > **W1**. In Section 3.3, this work states, “For other needed files, we prompt GPT-4o mini to extend the file content and generate Python code to produce the files.” However, the methodology for file generation remains unclear. For example, if a file is required to be an MP4 video of Taylor Swift’s latest album, it’s uncertain how this content could be generated through Python code alone. Furthermore, if GPT-4o mini generates the Python code to produce such files, it raises concerns about data quality and how the model ensures that the generated content is not hallucinated.
>
> **R:** In this work, we do not generate video files (e.g., MP4, MOV), and the current T3-Agent is not designed to handle video-based tasks. The primary reason is the lack of sufficiently reliable video tools. While we have explored video tasks, we found that even the most advanced tools (e.g., Gemini 1.5 Pro) struggle with challenges requiring fine-grained video understanding, complex video reasoning over long video sequences, and deep semantic interpretation.
>
> In future work, we will address the challenges by exploring more advanced video tools and integrating them into the T3-Agent. We also acknowledge that hallucination remains a significant challenge. To address this, rather than using GPT-4o to generate Python codes for video creation, we will adopt a retrieval-based strategy (similar to the generation strategy of image tasks) to mitigate such risks in video tasks.
>
>
> > **W2**. While including a human study to assess dataset quality is commendable, having only five experienced raters for a subset of the data may be too limited, potentially introducing biases based on individual preferences. Gathering feedback from a larger pool of participants, even with fewer data points per person, could strengthen claims about the dataset's effectiveness. Additionally, comparing MM-Traj to filtered-out data may not yield meaningful insight. Instead, comparisons with other established tool-usage datasets would likely provide more meaningful insights.
>
> **R:** Thanks for your comments. We add a more comprehensive user study during rebuttal, where we recruit 30 persons with rich programming and AI experience to evaluate the generated tasks and corresponding trajectories. Each person is tasked with assessing 20 tasks with trajectories, which are randomly selected and mixed from both MM-Traj and filtered-out data. We ask each one to provide scores (1-10) for the task quality and trajectory quality, and they do not know whether the data is from MM-Traj or filtered-out data. A low score means the quality is bad and a high score means its quality is good. The average scores are shown in Table J. The results clearly show that the used verifiers remove low-quality data. Due to the time limitation, we will further expand the scale of the user study in a revised version.
>
> **Table J: User study on the MM-Traj dataset**
> | MM-Traj Task | MM-Traj  Trajectory | Filtered out Task | Filtered out Trajectory |
> |-------------------|-------------------|-------------------|-------------------|
> |8.32|8.67|6.36|6.38|
>
> Thank you for your suggestion to compare MM-Traj with other established tool-usage datasets. However, it is difficult to make direct comparisons, due to the domain, complexity, and evaluation differences among these datasets. In the future, we will bridge this gap by introducing a unified inference formulation among these datasets and further conducting fair comparisons.

---

> > ### Author Response · Authors · 2024-11-20
> > **Response To Reviewer snR1 [2/2]**
> >
> > > **W3**. The evaluation results in Table 3 reveal mixed outcomes, with the T3-agent performing significantly worse than other methods, such as HF Agent and Sibyl Agent, on the GAIA benchmark. What could lead to this performance discrepancy on the GAIA benchmark?
> >
> > **R:** The GAIA benchmark is extremely challenging, since it has more complex tasks and longer trajectories, requiring stronger context understanding and reasoning capabilities. It is worth noting that the HF Agent and Sibyl Agent use the GPT-4o model as the controller, which is a closed-source model benefitting from larger-scale training data and larger model size. In contrast, the T3-Agent is developed with the MiniCPM-V model which is an 8.5B open-source model and fine-tuned using 20K data, which are much smaller than the model size and data volume of GPT-4o. Thus, it is not fair to directly compare the T3-Agent with these agents.
> >
> > To make comprehensive comparisons on the GAIA dataset, we compare the T3-Agent with agents using open-source models, including LLaVA-NeXT-8B, InternVL2-8B, and Qwen2-VL-7B, as shown in Table K. The T3-agent has obviously better performance than these agents. This experiment demonstrates the effectiveness of our method. We will add the above discussion and experiments in a revised version.
> >
> >
> > **Table K: Comparisons on the GAIA benchmarks with open-source models**
> > | Method | Controller | AnsAcc | Level1 | Level 2 | Level 3 |
> > |-------------------|-------------------|-------------------|-------------------|-------------------|-------------------|
> > |HF Agent|LLaVA-NeXT-8B|3.64|9.43|1.16|0.00|
> > |HF Agent|InternVL2-8B|4.85|7.55|4.65|0.00|
> > |HF Agent|Qwen2-VL-7B|9.70|16.98|8.14|0.00|
> > |HF Agent|MiniCPM-V-8.5B|7.27|13.21|5.81|0.00|
> > |T3 Agent|Tuned MiniCPM-V-8.5B|**15.15**|**26.42**|**11.63**|**3.84**|
> >
> >
> > > **Q1**. In Figure 3, the sum of trajectories—214 + 14,273 + 8,740 + 2,520 + 1,242 + 697 + 199 + 202—totals 28,087, which exceeds the stated 20k tasks in the abstract. In addition, the paper mentions that only 15k files remain after passing the query-file and trajectory verifiers. What is the final size of the generated dataset?
> >
> > **R:** Thank you for pointing this out. We put an incorrect Figure 3(d), where some data was inadvertently duplicated in the statistics. We correct it in the revised version. After double-checking, we confirm that there are a total of 20K tasks with 15K files after the two verifiers. This discrepancy between '20K' and '15K' arises because some tasks do not have assigned files and instead, they require searching for information from external web sources.
> >
> >
> >
> >
> > > **Q2**. Why does GPT-4o mini outperform GPT-4o in Table 2, specifically in the row with HF Agents on AnsACC and CodeExec, given that GPT-4o is expected to be more powerful than GPT-4o mini?
> >
> > **R:** The reason for the performance discrepancy is that GPT-4o sometimes uses its own knowledge to directly answer the question instead of adhering to the format of "Thought: ..., Code: ..." to call tools. This causes the code parsing error and slightly inferior performance to GPT-4o-mini.

---

> > > ### Comment · Reviewer_snR1 · 2024-11-26
> > >
> > > Thanks for the clarifications. I will raise my scores.

---

> > > > ### Author Response · Authors · 2024-11-27
> > > > **Thanks for your review!**
> > > >
> > > > Thank you for raising the score! We are delighted to hear that our response could clarify your concerns. Your invaluable feedback has been instrumental in enhancing this work. We deeply appreciate your thoughtful and constructive comments!

---

### Official Review · Reviewer_zCvz · 2024-11-01

**Soundness:** 3
**Presentation:** 3
**Contribution:** 3
**Rating:** 8
**Confidence:** 3

**Summary:**

This paper introduces a new approach for improving tool usage in multi-modal agents by fine-tuning a vision-language model (VLM) controller with synthesized tool-usage data. To overcome the limitations of traditional LLM-driven agents—such as reliance on prompt engineering and limited reasoning for tool usage across modalities—the authors create a three-step data synthesis pipeline. First, *Query Generation* uses GPT-4o mini to generate diverse, tool-aware prompts. Next, *File Generation* retrieves images from similar datasets and creates other files programmatically. Finally, *Trajectory Generation* employs a zero-shot agent using GPT-4o mini to solve tasks, capturing thought processes, code, and observations for each step. Quality is controlled through query-file and trajectory verifiers, also based on GPT-4o mini, producing a dataset called MM-Traj. The resulting agent, T3-Agent, uses the ReAct framework and the MiniCPM-V model trained on MM-Traj, enabling versatile tool usage across categories like web search, visual perception, image editing, and file handling. Benchmarks on GTA and GAIA demonstrate the T3-Agent’s significant improvements in tool usage over both untrained VLMs and other state-of-the-art LLM-driven agents.

**Strengths:**

- The data synthesis pipeline introduces a scalable, automated approach to generating diverse, complex multi-modal data for tool usage scenarios.
- Verification mechanisms embedded within the pipeline enhance data quality, resulting in a robust, comprehensive dataset.
- With training on the MM-Traj dataset, the T3-Agent demonstrates significant performance gains, surpassing agents built on closed-source models like GPT-4 in certain benchmarks.
- Ablation studies underscore the critical role of data verification in achieving top performance.
- The paper includes detailed visualizations of the T3-Agent’s reasoning process.

**Weaknesses:**

- The T3-Agent exhibits a gap in programming capabilities, which leads to lower accuracy in predicted answers compared to GPT-4o.
- While the paper acknowledges the T3-Agent’s limited programming capabilities, it does not suggest potential improvements or outline future directions to strengthen this aspect.
- The reliance on GPT-4o mini throughout the pipeline raises questions about biases and limitations from this closed-source model. Exploring alternative methods or open-source models could enhance transparency and address these limitations.
- What is $p_i$ in Equation 2?

**Questions:**

I lean towards acceptance because the paper introduces a promising approach to enhancing the reasoning capabilities of multi-modal agents. The proposed data synthesis pipeline and the resulting MM-Traj dataset are significant contributions to the field, potentially advancing the state of multi-modal learning. However, the paper lacks a discussion on the T3-Agent's robust programming capabilities and does not explore how these might be improved. I would like the authors to comment on this aspect.

---

> ### Author Response · Authors · 2024-11-20
> **Response To Reviewer zCvz**
>
> Thanks for your insightful and thorough review. We will address your concerns below.
>
>
> > **W1**. The T3-Agent exhibits a gap in programming capabilities, which leads to lower accuracy in predicted answers compared to GPT-4o.
>
> **R:** The performance gap in programming can likely be attributed to the differences in model size and training data of the agent controller.
> Compared with the MiniCPM-V model in the T3-Agent,  GPT-4o benefits from a larger model size and richer training corpus.
>
>
> > **W2**. While the paper acknowledges the T3-Agent’s limited programming capabilities, it does not suggest potential improvements or outline future directions to strengthen this aspect.
>
> **R:** Thanks for your comment. In the future, we could improve the programming capabilities of the agent via two manners. (1) Use VLMs that are pre-trained for coding as the controller. (2) Combine code data (such as the PyTraceBugs dataset [d]) with our trajectory data in fine-tuning.
>
> [d] PyTraceBugs: A Large Python Code Dataset for Supervised Machine Learning in Software Defect Prediction
>
>
> > **W3**. The reliance on GPT-4o mini throughout the pipeline raises questions about biases and limitations from this closed-source model. Exploring alternative methods or open-source models could enhance transparency and address these limitations.
>
> **R:** We chose GPT-4o mini due to its well recognition within the research community. However, we emphasize that our method is not tied to specific models and can flexibly transfer to open-source alternatives. To demonstrate this point, we add an experiment where the GPT-4o mini model in the data synthesis pipeline is replaced by the open-source Qwen2.5-72B model. Using this setup, we collect 12K new data and fine-tune the MiniCPM-V-8.5B model. The resulting performance is evaluated on the GAIA dataset, with the results presented in Table I. These results demonstrate that the 12K data produces about 6% improvements over the untuned baseline, highlighting the model-agnostic nature of our approach. We will include these findings in the revised manuscript to emphasize the flexibility and compatibility of our method with open-source models.
>
>
> **Table I: Agent on the GAIA benchmark.**
> | Data num | models in data synthesis | AnsAcc |Level 1 | Level 2 | Level 3 |
> |-------------------|-------------------|-------------------|-------------------|-------------------|-------------------|
> |Untuned agent| N/A | 7.27 | 13.21 | 5.81 | 0.00 |
> |12K| Qwen2.5-72B | 13.94 | 24.53 | 11.63 | 0.00 |
> |20K| GPT-4o-mini | 15.15 | 26.42 | 11.63 | 3.84 |
>
>
> > **W4**. What is $p_i$ in Equation 2?
>
> **R:** Thanks for pointing it out. $p_i$ should be corrected as $t_i$, denoting the generated thought in the $i$-th step, and the Equation 2 is
>
> \begin{equation}
> \min \mathbb{E}_{(F^{\star}, Q, T, C, O, A)\sim \mathbb{D} } [ -\sum^{n}_i P(t_i , c_i| F^{\star}, Q, h_i)],
> \end{equation}
>
> where we train the agent controller to fit the thought $t_i$ and code $c_i$. The equation will be revised in a new version.
>
> > **Q1**. I lean towards acceptance because the paper introduces a promising approach to enhancing the reasoning capabilities of multi-modal agents. The proposed data synthesis pipeline and the resulting MM-Traj dataset are significant contributions to the field, potentially advancing the state of multi-modal learning. However, the paper lacks a discussion on the T3-Agent's robust programming capabilities and does not explore how these might be improved. I would like the authors to comment on this aspect.
>
> **R:** Thanks for your positive comment. To improve the programming capabilities of agents, there are two manners: (1) using VLMs pre-trained for coding and (2) combining code data with our trajectory data in fine-tuning. We will add the above discussion in the revised manuscript.

---

> > ### Comment · Reviewer_zCvz · 2024-11-29
> >
> > Thank you for the updated paper! My concerns have been addressed. I will maintain my positive score.

---

> > > ### Author Response · Authors · 2024-11-29
> > > **Thanks for your review!**
> > >
> > > Thank you for your thoughtful review! We are pleased to hear that our response addressed your concerns. Your invaluable feedback has played a crucial role in improving this work!

---

### Official Review · Reviewer_uszN · 2024-11-02

**Soundness:** 3
**Presentation:** 4
**Contribution:** 3
**Rating:** 8
**Confidence:** 3

**Summary:**

This paper presents a novel approach for multi-modal agent tuning, aimed at enhancing agent performance in complex environments through better utilization of multiple data modalities. The authors propose a tuning framework designed to leverage cross-modal data (e.g., visual and text info) to improve agent task performance, with specific emphasis on tool usage within the agent's capabilities. Their T3 agent is multi-modal agent that can efficiently use tools to solve practical tasks by tuning a VLM as the controller. Evaluations over the various dataset show the significant improvement using their agent with closed as well as open source model. Additionally, they curated dataset using multimodal info having trajectory of various length for broader study. In this work focus is on the correct tool selection and code is given more importance than the widely used JSON schema. In summary, they generate data, tune the VLM and create dataset followed by leverage of tool agent to make use of tool in a better way.

**Strengths:**

1. Innovation in Multi-Modal Interaction: The approach shows potential in pushing the boundaries of how agents interact with cross-modal data sources. By focusing on practical applications of tool usage, this work could offer useful insights into building agents that understand and respond to complex queries across various media.
2. Comprehensive Methodology: The paper describes a well-structured experimental setup and provides a clear description of the multi-modal tuning process. This includes thoughtful considerations on data processing, model architecture, and task-specific tuning steps, making it easy to follow.
3. Evaluation Metrics: The authors employ a diverse set of metrics to evaluate agent performance. This choice not only validates the model’s accuracy in the tasks but also emphasizes the practical utility of the proposed framework in real-world applications.

**Weaknesses:**

1. Interpretability: While the approach demonstrates performance gains, the interpretability of results remains limited. Additional analyses, such as ablation studies or attention maps, would be beneficial to understand how each modality contributes to the decision-making process. For example paper first generate queries first without files before relevant queries, what is the impact if we don't do that, or this is based on some past work/observations?

2. Scalability: The paper does not thoroughly address the scalability of the proposed method, particularly as the number of modalities or the dataset size increases. It would be beneficial to test how the method's performance and computational requirements scale with additional modalities or larger datasets. For example, experiments that measure latency, memory usage, and accuracy as more data is introduced could illustrate the framework's robustness and its viability in resource-constrained or high-throughput environments.

3. User Study: To evaluate the practical usability of the framework, a small user study or qualitative feedback from users would provide valuable insights into the query handling experience. Specifically, gathering feedback on aspects like ease of use, perceived accuracy, responsiveness to complex queries, and the intuitiveness of the tool-usage process could highlight areas for refinement in real-world settings.

Minor hints:
1. sec5.6 typo.. "wen based"--> web based.
2. sec 3.4- Author(s) mention details can be found in but missed cross-referencing it.
3. cross reference missing at end of sec 3.4

**Questions:**

Q: Could you please elaborate on how not using the final answer A aligns with your goal? Specifically, how does this choice benefit your approach to enhancing tool-usage capability?

---

> ### Author Response · Authors · 2024-11-20
> **Response To Reviewer uszN [1/2]**
>
> Thanks for your insightful and thorough review. We will address your concerns below.
>
> > **W1**. Interpretability: While the approach demonstrates performance gains, the interpretability of results remains limited. Additional analyses, such as ablation studies or attention maps, would be beneficial to understand how each modality contributes to the decision-making process. For example paper first generates queries first without files before relevant queries, what is the impact if we don't do that, or this is based on some past work/observations?
>
> **R:** Thanks for your comments. For the interpretability of our method, we add ablation experiments to show the contributions of different modalities in decision-making. Ablation results on the GTA dataset are shown in Table E, where removing the image modality reduces the performance by 40%, highlighting the importance of input images.
>
> **Table E: Ablation on the GTA Benchmark**
> | Method | AnsAcc | ToolAcc | ToolAcc |
> |-------|------|---|---|
> |T3 Agent w/o image|10.67 | 25.32 | 20.09 |
> |T3 Agent w/ image | **52.56** | **65.85** | **80.49** |
>
>
> Regarding your concern "first generate queries without files before relevant queries", we adopted this strategy based on our observations, as it leads to more natural synthetic tasks. For complex tasks involving multiple files, first generating files and then queries often result in less natural tasks, as the generated files within one query may have limited or unrelated connections. By generating queries first and then creating files accordingly, we can ensure stronger coherence and relevance among the files based on the information in queries.
>
> > **W2**. Scalability: The paper does not thoroughly address the scalability of the proposed method, particularly as the number of modalities or the dataset size increases. It would be beneficial to test how the method's performance and computational requirements scale with additional modalities or larger datasets. For example, experiments that measure latency, memory usage, and accuracy as more data is introduced could illustrate the framework's robustness and its viability in resource-constrained or high-throughput environments.
>
> **R:** For the scalability of modalities, our method is able to extend to additional modalities by incorporating more tools and leveraging advanced multimodal models.  For example, to extend our method to the video modality, we can integrate a video search model into the data synthesis pipeline and replace the MiniCPM-V model with a video-language model for the agent. This approach ensures seamless adaptation to new modalities while maintaining efficiency and coherence.
>
> For the scalability of dataset size, we add experiments in Table F to show the agent's performance on the GTA benchmark as the dataset size increases. With the increase of data number, the agent achieves better performance, the memory consumption is constant, and the time consumption linear increases. Compared with the accuracy improvements, we think that the consumption of memory and time is acceptable.
>
> **Table F: Ablation on the GTA Benchmark**
> | Dataset size | 6K | 12K | 20K |
> |--|--|--|--|
> |Accuracy|43.59%|48.08%|52.56%|
> |Memory|214 GB|214 GB|214 GB|214 GB|
> |Training Time|276 mins|532 mins|946 mins|
>
> We agree with the reviewer about the importance of the scalability of modalities and dataset size. However, our current work mainly focuses on demonstrating the effectiveness of the multimodal agent tuning method. We will expand to additional modalities and explore engineering optimizations in the future, to enhance the applicability of our method by using fewer resources but for more tasks.
>
> > **W3**. User Study: To evaluate the practical usability of the framework, a small user study or qualitative feedback from users would provide valuable insights into the query handling experience. Specifically, gathering feedback on aspects like ease of use, perceived accuracy, responsiveness to complex queries, and the intuitiveness of the tool-usage process could highlight areas for refinement in real-world settings.
>
> **R:** Thanks for your comments. We add a user study about agent outputs on the GTA benchmark. We recruited 20 participants, each evaluating 20 tasks with agent outputs, where the agent is with or without fine-tuning. The outputs of the agent (w/ or w/o tuning) were shuffled for each task, and the participants were not informed about the source, ensuring an unbiased assessment. The participants were asked to provide the preference of the two agent outputs, based on the accuracy, helpfulness, and relevance. We measured the percentages of results, as shown in Table G. Outputs from the tuned agent has a significantly high preference, indicating its better performance in solving practical tasks.
>
> **Table G: User study for agent outputs on the GTA benchmark.**
> | Agent w/o tuning is better | Tie | Agent w tuning is better|
> |--|---|--|
> |21%|13%|66%|

---

> > ### Author Response · Authors · 2024-11-20
> > **Response To Reviewer uszN [2/2]**
> >
> > > **Q1**. Could you please elaborate on how not using the final answer $A$ aligns with your goal? Specifically, how does this choice benefit your approach to enhancing tool-usage capability?
> >
> > **R:** We do not use the final answer $A$ in the training objective of the agent controller, as we encourage the controller to leverage tools in solving given tasks, instead of directly producing an answer based on its internal knowledge.
> >
> > > **Minor hints**
> > > 1. sec5.6 typo.. "wen based"--> web based.
> > > 2. sec 3.4- Author(s) mention details can be found in but missed cross-referencing it.
> > > 3. cross reference missing at end of sec 3.4
> >
> > **R:** Thank you again for your detailed feedback and valuable suggestions. We have corrected these typos and conducted a thorough review of the manuscript to ensure that no other typos remain.

---

> > ### Comment · Reviewer_uszN · 2024-11-21
> > **Response to Authors comment**
> >
> > Thank you for addressing my concern and I am happy to see that there is a constant memory use even with increasing the dataset size. Regarding W3 (user study), what are the details of these 20 participants, as in the paper I see that 5 persons were used to evaluate the tasks and trajectories generated. What is the reason for this difference?  Are these participants familiar with the whole setup beforehand? Suppose the evaluators are all from the same lab as the authors this has a potential of creating bias in user study.
> >
> > Apart from user study my other concerns are addressed.

---

> > > ### Author Response · Authors · 2024-11-21
> > > **Response To Reviewer uszN**
> > >
> > > > Thank you for addressing my concern and I am happy to see that there is a constant memory use even with increasing the dataset size.
> > >
> > > > Regarding W3 (user study), what are the details of these 20 participants?
> > >
> > > Thank you for your response. We are also glad to see that your other concerns have been addressed.
> > >
> > > Regarding the 20 participants, they are from multiple universities and research institutions (not in the same lab). While they are all members of the AI community, they come from different research backgrounds and are not familiar with our project beforehand.
> > >
> > > The criteria for selecting participants were as follows:
> > >
> > > - Participants are required to have a certain level of programming proficiency, as the controller's output consists of Thought and Python Code. Assessing the quality of the Python Code necessitates programming skills.
> > > - Participants are unaware of our method, including the data synthesis pipeline and the agent architecture. We only introduce the structure of the trajectories and the meaning of the agent outputs, expecting the participants to evaluate the quality of trajectories and agent outputs objectively.
> > >
> > > > As in the paper I see that 5 persons were used to evaluate the tasks and trajectories generated. What is the reason for this difference?
> > >
> > > **R:** The 5 participants mentioned in the paper (increased to 30 during the rebuttal period) and the 20 participants referenced here are not the same group. This distinction is made to avoid potential biases in evaluations, such as additional priors about our agent.
> > >
> > > The former group primarily evaluated the quality of generated tasks and trajectories via our data synthesis pipeline, while the latter assessed the execution results of the agent.
> > >
> > > > Are these participants familiar with the whole setup beforehand?
> > >
> > > **R:** No, the participants are not familiar with the setup beforehand. We only informed them about the agent's current functionality and asked them to rate its performance based on their preferences in terms of accuracy, helpfulness, and relevance. We did not disclose any details about the implementation of our agent or the data synthesis process to avoid evaluation biases caused by prior knowledge.
> > >
> > > > Suppose the evaluators are all from the same lab as the authors this has a potential of creating bias in user study.
> > >
> > > **R:** No. They are not from the same lab to reduce the biases in user study. Our participants are from multiple universities and research institutions.

---

> > > > ### Comment · Reviewer_uszN · 2024-11-28
> > > > **Response to authors comment (round 2)**
> > > >
> > > > Thank you for your responses. My queries have been resolved, and with more participants in the user study, the findings are clearer for the readers. I will maintain my rating and recommend accepting the paper.

---

> > > > > ### Author Response · Authors · 2024-11-28
> > > > > **Thanks for your review!**
> > > > >
> > > > > Thank you for your thoughtful review and positive feedback! We sincerely appreciate your valuable suggestions on interpretability, scalability, and the user study, which have been instrumental in enhancing this work!

---

### Official Review · Reviewer_4Mmx · 2024-11-05

**Soundness:** 3
**Presentation:** 3
**Contribution:** 3
**Rating:** 6
**Confidence:** 4

**Summary:**

This paper proposes a method for multi-modal agent tuning for tool usage and presents a dataset designed to train an agent for tool usage. The authors claim that their T3-Agent achieves significant improvements.

**Strengths:**

1) The idea of using an LLM to separately generate queries, files, and trajectories, followed by a query-file verifier and trajectory verifier is neat.
2) The paper addresses the problem of using the appropriate array of tools corresponding to the information relevant to a query well.
3) The experiments are thorough.

**Weaknesses:**

1) Verifying the output of an LLM by the LLM itself does not seem accurate. I am skeptical about the quality of the generated MM-Traj dataset.
2) You need quantitative verification for the dataset. (It is not clear whether a user study involving a few people on 100 data points out of 15K would provide sufficient confidence.)
3) Experimental results on the GTA benchmark are more promising than those on the GAIA benchmark. However, the overall performance of T3-agent is not superior to that of the other agents in comparison. Specifically, on the GAIA benchmark, the HF agent with GPT-4o performs twice as well as the T3-Agent.
4) If the T3-Agent’s performance were clearly superior, I would be more optimistic about the generated dataset. However, the results seem to support my doubts about the dataset.

Minor comment: In Tables 2 and 3, the best results should be highlighted in bold.

**Questions:**

Please address the concerns raised in the weaknesses.

---

> ### Author Response · Authors · 2024-11-20
> **Response To Reviewer 4Mmx [1/2]**
>
> Thanks for your insightful and thorough review. We will address your concerns one by one.
>
> > **W1**. Verifying the output of an LLM by the LLM itself does not seem accurate. I am skeptical about the quality of the generated MM-Traj dataset.
>
> **R:** Using LLMs to verify the quality of LLM outputs has shown effectiveness in multiple works, such as [a] verifying the generated instructions and [b] verifying the generated plans. Inspired by them, we argue that using LLMs can also verify the synthetic tasks and trajectories. The ablation experiments about the used verifiers are shown in Table 5, where the verifiers lead to about 2% improvements on both GTA and GAIA benchmarks. Meanwhile, a more solid user study (involving more people and more data points) is presented in the response to W2, which also confirms that the verifiers can filter our low-quality data.
>
>
> To evaluate the quality of the remaining data in the MM-Traj dataset, we have compared models with and without using the MM-Traj dataset, as shown in Table 2 and Table 3 of the manuscript. Using our dataset leads to 18.59% and 7.88% improvements on the GTA and GAIA benchmarks, respectively, demonstrating the effectiveness of the MM-Traj dataset.
>
>
> [a] Self-Instruct: Aligning Language Models with Self-Generated Instructions. ACL 2023.
>
> [b] APIGen: Automated Pipeline for Generating Verifiable and Diverse Function-Calling Datasets. NeurIPS 2024.
>
>
>
> > **W2**. You need quantitative verification for the dataset. (It is not clear whether a user study involving a few people on 100 data points out of 15K would provide sufficient confidence.)
>
>
> **R:** To address your concern, we conduct a more extensive and robust user study during rebuttal. Specifically, we recruited 30 participants with rich experience in programming and AI to evaluate the quality of the generated tasks and trajectories. Each participant was tasked with assessing 20 tasks and 20 trajectories, randomly selected and mixed from both the MM-Traj dataset and the filtered-out data. There are 600 tasks and 600 trajectories in total. Notably, the participants were blinded to the source of the data to ensure unbiased evaluations. Participants rated the quality of tasks and trajectories on a range from 1 to 10, where a higher score indicates better quality. The average scores, as shown in Table A, show that the MM-Traj dataset significantly outperforms the filtered-out data in both task and trajectory quality. These results confirm the effectiveness of our verification process.
>
>
>
>
> **Table A: User study for the verification process, where 30 persons were recruited and each one was assigned with 20 tasks and 20 trajectories.**
> | MM-Traj Task | MM-Traj  Trajectory | Filtered out Task | Filtered out Trajectory |
> |--|-|-|-|
> |8.32|8.67|6.36|6.38|
>
>
> > **W3**. Experimental results on the GTA benchmark are more promising than those on the GAIA benchmark. However, the overall performance of T3-agent is not superior to that of the other agents in comparison. Specifically, on the GAIA benchmark, the HF agent with GPT-4o performs twice as well as the T3-Agent.
>
> **R:** Compared to the GTA benchmark, the GAIA benchmark is more challenging due to its more complex tasks and longer trajectories, which demand stronger context understanding and reasoning capabilities. Although the HF agent using GPT-4o achieves higher performance than ours, it is important to note that GPT-4o benefits from significantly larger-scale training data and larger model size. In contrast, our agent is built on the MiniCPM-V-8.5B model, an open-source model with 8.5B parameters, fine-tuned on the MM-Traj dataset of 20K samples. The scales are considerably smaller than those of GPT-4o in both model size and data volume, which understandably lead to performance differences. Therefore, directly comparing our agent with agents using GPT-4o is not entirely fair.
>
> To provide more comprehensive and fair comparisons on the GAIA benchmark, we include results from agents built with open-source models of similar sizes, such as LLaVA-NeXT-8B, InterVL2-8B, and Qwen2-VL-7B, as shown in Table B. These comparisons demonstrate the competitive performance of T3-Agent within the open-source ecosystem. We will add the comparisons in a revised version.
>
> **Table B: Comparisons on the GAIA benchmarks with open-source models**
> | Method | Controller | AnsAcc | Level 1 | Level 2 | Level 3 |
> |--|-|-|-|--|-|
> |HF Agent|LLaVA-NeXT-8B|3.64|9.43|1.16|0.00|
> |HF Agent|InternVL2-8B|4.85|7.55|4.65|0.00|
> |HF Agent|Qwen2-VL-7B|9.70|16.98|8.14|0.00|
> |HF Agent|MiniCPM-V-8.5B|7.27|13.21|5.81|0.00|
> |T3 Agent|Tuned MiniCPM-V-8.5B|**15.15**|**26.42**|**11.63**|**3.84**|

---

> > ### Author Response · Authors · 2024-11-20
> > **Response To Reviewer 4Mmx [2/2]**
> >
> > > **W4**. If the T3-Agent’s performance were clearly superior, I would be more optimistic about the generated dataset. However, the results seem to support my doubts about the dataset.
> >
> > **R:** In fact, generating high-quality data for multimodal agents is extremely challenging due to the requirements for naturalness, diversity, and complexity of both the tasks and their corresponding trajectories. Our paper proposes an effective pipeline for data generation, resulting in the MM-Traj dataset. Although the T3-Agent does not outperform all agents derived from closed-source models, this does not imply that the MM-Traj dataset is of low quality. Below are the supporting arguments.
> >
> >
> > (1) We have compared the MiniCPM-V model with and without using the MM-Traj dataset, as shown in Table 2 and Table 3. Using our dataset yields improvements of 18.59% and 7.88% on the GTA and GAIA benchmarks, respectively, highlighting the effectiveness of both the MM-Traj dataset and the proposed data synthesis pipeline.
> >
> > (2) We add a more extensive and robust user study that shows the employed verifiers can indeed discard low-quality data. (Details in the response to W2)
> >
> >
> > (3) These agents that achieve better performance than the T3-Agent are based on closed-source models (e.g., GPT-4o) with larger model size and more training data. These factors may primarily contribute to the performance differences. To make more comprehensive comparisons, we compare the T3-Agent with agents driven by open-source models (details are shown in the response to W3). T3-Agent achieves better performance, underscoring the effectiveness of the MM-Traj dataset.
> >
> >
> > (4) We evaluate the MM-Traj datasets on more VLMs. Concretely, we use the MM-Traj dataset to fine-tune another VLM: Qwen2-VL-7B. Results on the GTA and GAIA benchmarks are shown in Table C and Table D, respectively. Compared with the untuned Qwen2-VL-7B model, the tuned model brings 19% and 7% improvements on the two benchmarks. The performance improvements across **multiple VLMs** further validate the effectiveness of our dataset.
> >
> >
> > We will include the above discussion and experiments in the revised version.
> >
> > **Table C: Comparisons on the GTA dataset**
> > | Controller | AnsAcc | ToolAcc | CodeExec |
> > |-------------------|-------------------|-------------------|-------------------|
> > |MiniCPM-V-8.5B|33.97|36.59|56.10|
> > | Tuned MiniCPM-V-8.5B | **52.56**  | **65.85**  | **80.49**  |
> > | Qwen2-VL-7B | 42.31 | 44.85 | 65.19|
> > | Tuned Qwen2-VL-7B | **53.85**  |  **64.63**  | **84.32**  |
> >
> >
> > **Table D: Comparisons on the GAIA dataset**
> > | Controller | AnsAcc | Level 1 | Level 2 | Level 3 |
> > |-------------------|-------------------|-------------------|-------------------|-------------------|
> >  |MiniCPM-V-8.5B|7.27 | 13.21 | 5.81 | 0.0 |
> >  | Tuned MiniCPM-V-8.5B | **15.15** | **26.42** | **11.63** | **3.84** |
> >  | Qwen2-VL-7B |9.70|16.98|8.14|0.00|
> >  | Tuned Qwen2-VL-7B | **16.97** | **26.42** | **15.12** | **3.84** |

---

> > > ### Comment · Reviewer_4Mmx · 2024-11-27
> > > **Concerns addressed**
> > >
> > > Thanks for your comments, I will raise my score.

---

> > > > ### Author Response · Authors · 2024-11-28
> > > > **Thanks for your review!**
> > > >
> > > > Thank you for your thoughtful review and for raising the score! We’re thrilled to know that our response addressed your concerns effectively. Your constructive comments have been essential in improving this work, and we truly appreciate the suggestions you’ve provided.

---

### Author Response · Authors · 2024-11-26
**General Response**

Dear Area Chairs and Reviewers,

We sincerely thank the Area Chairs and all the Reviewers for their insightful comments and recognition of our work. Your feedback has been invaluable in helping us improve our paper. We are particularly grateful for the acknowledgment of the strengths of our work, including:


1. Our multi-modal agent tuning method is **neat** (Reviewer 4Mmx), **novel** (Reviewer uszN), **well-structured** (Reviewer uszN), and **scalable** (Reviewer zCvz).
2. Our data synthesis pipeline produces **diverse** and **expressive** tasks (Reviewer snR1).
3. The dataset is **robust**(Reviewer zCvz), **comprehensive**, (Reviewer zCvz) and **novel** (Reviewer snR1) with **in-depth statistical analysis** (Reviewer snR1).
4. The experiments are **thorough** (Reviewer 4Mmx), use **diverse valuation metrics** (Reviewer uszN), have **significant performance gains** (Reviewer zCvz) and **detailed visualizations** (Reviewer zCvz).

Based on your feedback, we have made revisions to our paper. Below is a summary of the major updates.

**1. We conduct several new experiments to strengthen our contributions.**

(1) We compare the T3-Agent with multiple agents driven by open-source models. Results show significant improvements in the GTA and GAIA benchmarks, validating the effectiveness of our dataset and data synthesis pipeline.

(2) We tune another model: Qwen2-VL-7B as the controller using the MM-Traj dataset, and it achieves consistent improvements on both benchmarks.

(3) We conduct ablation studies on input modalities and dataset sizes to evaluate their individual impacts.

(4) We conduct more comprehensive user studies for generated tasks, trajectories, and agent outputs. We recruit more people (30 (for data quality) + 20 (for agent outputs) = 50 persons in total) and assess the quality of more data points (1000 data points in total). Results show the effectiveness of the used verifiers and the obtained dataset.

(5) Replacing GPT-4o-mini with Qwen2.5-72B, we verify that our data synthesis pipeline remains effective, and the generated data can improve agent performance as well.

**2. We improve the presentation of our paper to make it more clear and readable.**

(1) We clarify that the two used verifiers effectively filter low-quality synthetic data, as demonstrated in prior works, our experiments, and user studies.

(2) We discuss that the performance gap between T3-Agent and GPT-4o-driven agents primarily arises from differences in model size and training data scale of the controller.

(3) We add explanations on how our method scales to more modalities by incorporating additional tools and advanced multi-modal models. Furthermore, synthesizing larger datasets consistently enhances performance.

(4) We fix typos and correct an erroneous figure for accuracy.

Thanks again to the Area Chairs and all the Reviewers! We look forward to any further discussions or questions during the rebuttal phase.

Best regards,

The Authors

---

### Meta-Review · Area_Chair_RsPK · 2024-12-17

**Metareview:**

The paper presents an approach to improve the tool-use capability of vision-language models using synthetic tool-use dataset. The pipeline for synthetic data consists of three steps. First, LLMs generate query tasks using a tool-aware prompt. Second, task relevant images are retrieved or generated. Finally, using the ReACT framework, trajectories containing steps, thought, and code are generated for each generated task. Finetuning a VLM on this dataset shows considerable improvements over off-the-shelf open-weights VLM on the GTA and GAIA benchmarks.

Reviewers agreed that the synthetic data generation pipeline proposed in the paper is novel and scalable. Having a verification mechanism integrated into the pipeline was also seen as a strength by the reviewers. Reviewers were also happy with the experimental setup, and significant performance gains on the benchmarks.

**Additional Comments On Reviewer Discussion:**

During the reviewer discussion period, all reviewers were satisfied with the authors responses. I am summarising the discussion here:

The authors conducted a more thorough user study (as requested by Reviewer 4Mmx, uszN, snR1) and found that trajectories and tasks that were filtered out scored lower than those that were part of the final dataset indicating the effectiveness of the proposed verification process.

The authors also provided more baselines of the same size as the proposed approach for the GAIA benchmark to put the work in perspective. Adding this to the final manuscript will be useful.

Reviewer 4MMx had concerns about the quality of generated dataset, and its effectiveness. In response, authors conducted additional experiments with Qwen2-VL along with MiniCPM, and finetuned versions on the synthetic dataset showed improvements on both GTA and GAIA showing the effectiveness of the dataset.

Authors should also add the discussion with uszN about reasons for improvement. Additional ablations showed in Table E and Table F could be added to the appendix.


It is also encouraging to see that the proposed data pipeline also works with open-weights model. The authors show that replacing GPT-4o-mini with Qwen-2.5.7B during synthetic data pipeline also leads to improvement (albeit a smaller improvement).

---

### Decision · Program_Chairs · 2025-01-22

Accept (Spotlight)